# Lipopolysaccharide binding protein resists hepatic oxidative stress by regulating lipid droplet homeostasis

Qilun Zhang[1,14], Xuting Shen[2,14], Xin Yuan[2,14], Jing Huang[2], Yaling Zhu[2], Tengteng Zhu[2], Tao Zhang[2], Haibo Wu[3], Qian Wu[4], Yinguang Fan[5], Jing Ni[5], Leilei Meng[2], Anyuan He[6], Chaowei Shi ®[7], Hao Li[7], Qingsong Hu[8], Jian Wang ®[9], Cheng Chang ®[9], Fan Huang[10], Fang Li[11], Meng Chen[12], Anding Liu ®[13] ✉, Shandong Ye[1] ✉, Mao Zheng[1] ✉ & Haoshu Fang ®[2] ✉

Oxidative stress-induced lipid accumulation is mediated by lipid droplets (LDs) homeostasis, which sequester vulnerable unsaturated triglycerides into LDs to prevent further peroxidation. Here we identify the upregulation of lipopolysaccharide-binding protein (LBP) and its trafficking through LDs as a mechanism for modulating LD homeostasis in response to oxidative stress. Our results suggest that LBP induces lipid accumulation by controlling lipid-redox homeostasis through its lipid-capture activity, sorting unsaturated tri-glycerides into LDs. N-acetyl-L-cysteine treatment reduces LBP-mediated tri-glycerides accumulation by phospholipid/triglycerides competition and Peroxiredoxin 4, a redox state sensor of LBP that regulates the shuttle of LBP from LDs. Furthermore, chronic stress upregulates LBP expression, leading to insulin resistance and obesity. Our findings contribute to the understanding of the role of LBP in regulating LD homeostasis and against cellular peroxidative injury. These insights could inform the development of redox-based therapies for alleviating oxidative stress-induced metabolic dysfunction.

Oxidative stress, resulting from an imbalance between reactive oxygen species (ROS) production and the antioxidant defense system, is a key driver of hepatic steatosis and obesity[1–3]. Toxic lipid metabolites such as diacylglycerols, lysophosphatidyl choline, and other unsaturated metabolites generated due to oxidative stress, cause cellular damage[4]. This lipotoxicity considered a potential pro-steatotic mediator, is also a key factor in the progression of metabolic dysfunction. Therefore, even in the absence of overweight, acute hunger or chronic stress-induced oxidative stress can cause severe disruptions in lipid metabolism, further increasing the risk of all-cause mortality[5]. While antioxidants and ROS scavengers can alleviate metabolic dysfunction in animal models and some antioxidants have been trialed in clinical settings, long-term antioxidant treatment has potential safety issues, and there is currently no ideal drug available[6,7]. Thus, it is crucial to

understand the molecular mechanisms that lead to oxidative stress-induced lipid metabolic dysfunction.

Recent studies have shown that lipid droplets (LDs), the major organelles responsible for neutral lipid accumulation, not only function in lipid storage and mobilization but also serve as an essential component of the cellular antioxidant system[8]. The homeostasis regulation of LD is of great significance for the development and survival of cells under stress[9]. During oxidative stress, the biogenesis of LDs is stimulated to protect vulnerable lipids such as unsaturated fatty acids (UFAs) rerouted into the triglyceride (TG) core from ROS-induced peroxidation to maintain lipid homeostasis[10–12], which is subsided when oxidative stress ends. This process is accurately regulated by LD-associated proteins. Despite the identification of numerous LD-associated proteins[13], the cellular

A full list of affiliations appears at the end of the paper. ✉e-mail: andingliu@tjh.tjmu.edu.cn; ysd196406@163.com; zhengmao1999@foxmail.com; fanghaoshu@ahmu.edu.cn

sorting mechanisms for UFA-TG in response to oxidative stress are poorly understood.

In this study, we aimed to elucidate a fundamental mechanism that regulates the oxidative stress response of LDs. We have uncovered a cytoprotective mechanism, in which lipopolysaccharide-binding protein (LBP) mediates the selective sequestration of UFA-TG into LDs, effectively preventing lipolysis under oxidative stress. This finding is consistent with the previously reported antioxidant effect of LBP[14,15]. Importantly, we report a cellular mechanism by which protein/TG interaction promotes LD accumulation, which is consistent with the previously reported phenotype of reduced liver fat accumulation in LBP knockout mice[16,17]. Mechanistically, we found that LBP colocalizes with TG in LDs, which is regulated by the TG capture activity. The shuttle of LBP with LDs is meticulously regulated by its interaction with peroxiredoxin 4 (PRDX4), which senses oxidative stress and ensures the execution of LBP antioxidant function. In summary, LBP acts as an antioxidant to control lipid homeostasis and defends against oxidative stress by coupling with redox signaling and lipid metabolism.

## Results

### Oxidative stress induces LBP upregulation and localization within LDs

To investigate the compositional and regulatory changes of LDs under oxidative stress, we conducted hepatic transcriptomic and LDs proteomic analyses of 8-week-old C57BL/6J mice before and after a 24-h fast (Fig. 1a). Fasting elicits a hepatic stress response, as evidenced by upregulation of Cyp and peroxisome protein families and down-regulation of RPS and EIF families (Supplementary Data 1). An integrative analysis of liver transcriptome and LDs proteome data unveiled seven proteins exhibiting significant alterations (Fig. 1b). The majority of the identified proteins are known to be involved in LDs metabolism, with the exception of LBP, an acute-phase protein that functions as an extracellular Lipopolysaccharide (LPS) ligand[18-20]. We validated the findings from multi-omics data and discovered that fasting significantly elevated hepatic LBP expression level and its localization to LDs, suggesting an upregulation of LD-localized LBP expression in response to this metabolic stress challenge (Fig. 1c). We found that the expression of LBP in HepG2 cells was not upregulated by treatment with unsaturated fatty acid analog Bodipy 558/568 C12 (Bodipy C12), saturated fatty acid palmitic acid (PA), or phosphatidylcholine (Fig. 1d). Therefore, the upregulation of LBP expression is not a direct consequence of fatty acid exposure. Instead, we identified that oxidative stress, induced by starvation, hydrogen peroxide ($H_2O_2$), and heat shock, can enhance LBP expression (Fig. 1e). Our findings revealed that these oxidative stresses not only upregulated the expression of LBP but also caused its translocation to the LDs with Bodipy C12 (Fig. 1f). These results suggest that oxidative stress plays a critical role in regulating both LBP expression and subcellular distribution.

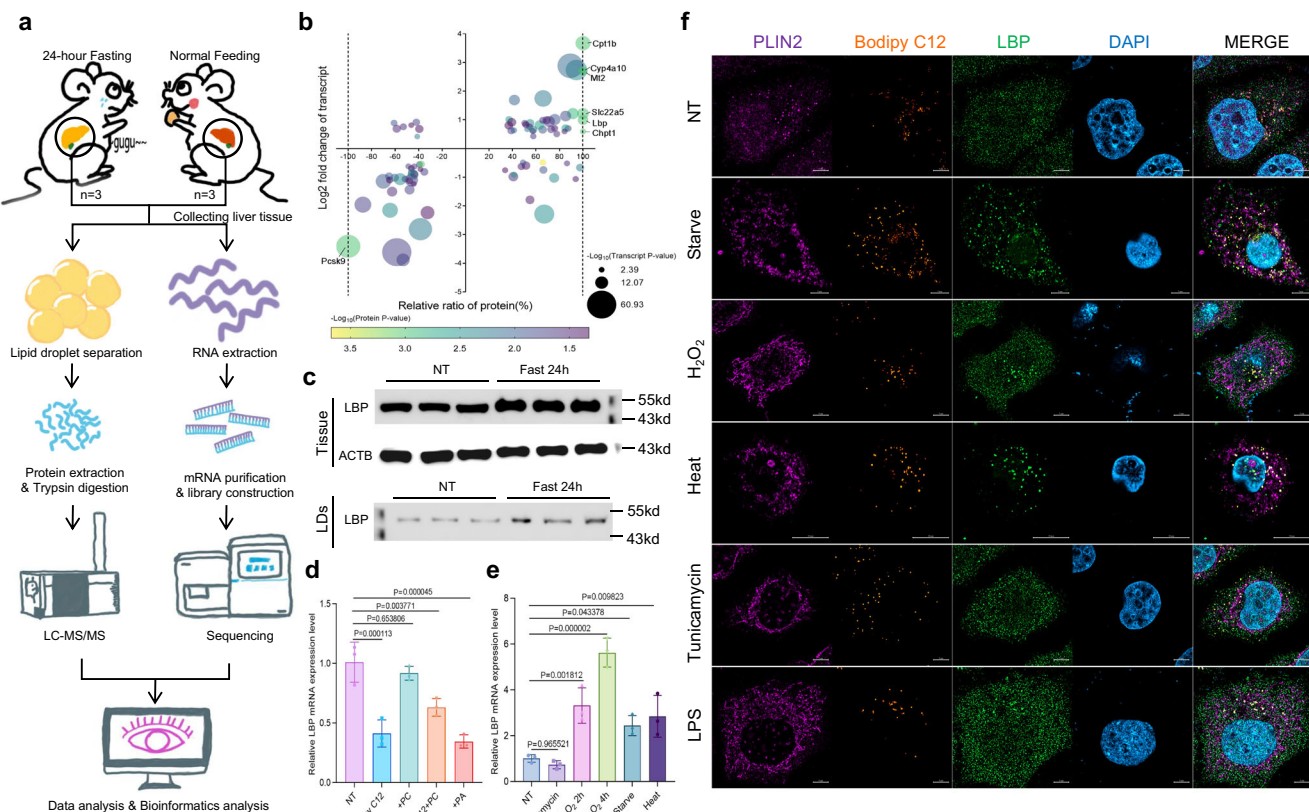

**Fig. 1 | Oxidative stress induces LBP upregulation and LD translocation. a** A schematic illustrating the research strategy adopted in this study. The image was created using materials supplied by Ge Pan from the Central Academy of Fine Arts, China, and illustrated by Qilun Zhang. The use of these materials has been authorized. **b** A multivariate bubble plot comparing changes in LD-localized protein and RNA expression levels after a 24-h fast. Two-tailed student's *t*-test (each group *n* = 3, biologically independent). **c** Western blot analysis of LBP in liver tissue or liver LDs from 8-week-old mice receiving no treatment (NT) or a 24-h fast. **d** Changes in LBP mRNA expression after treatment with 1 μM Bodipy C12, 2 μM PC, or 60 μM PA for 24 h. Shown are means ± s.d., one-way ANOVA (*n* = 3, biologically independent). **e** Changes in LBP mRNA expression after treatment of HepG2 cells with 4 μg/ml tunicamycin for 8 h, 1 mM hydrogen peroxide for 2 or 4 h, nutritional-starvation for 3 h, or 50 °C heat shock for 1 h. Shown are means±s.d., one-way ANOVA (*n* = 3, biologically independent). **f** Confocal microscope co-staining images of LBP (green), PLIN2 (violet), and DAPI (blue) captured 16 h after the treatment with Bodipy C12 (orange), 1 μg/ml LPS for 8 h, and the other treatments as described in (**e**). Scale bar = 5 μm. Experiments repeated three times independently with similar results.

## The localization of LBP in LDs leads to hepatocyte steatosis

Next, we investigated whether LBP was a causative factor for the formation of large LDs in hepatocytes. Fasting led to significant TG accumulation and steatosis, which was consistent with the intraperitoneal injection of the LBP inducer LPS (Fig. 2a, b). Additionally, LBP expression was also increased in HepG2 cells treated with $H_2O_2$, and this increase was positively correlated with Caveolin, an LD marker protein (Supplementary Fig. 1a). To assess the effects of LBP overexpression on LD formation, we transfected LBP-GFP into HepG2 cells and added Bodipy C12 to trace LD generation. We observed the formation of numerous large-sized droplets that co-localized with the LD marker Bodipy 665/676 (Supplementary Fig. 1b, c). Subsequently, we established animal models of conventional knockout (LBP$^{-/-}$) mice and Alb-LBP-3*flag (LBP$^{KI/KI}$) mice to investigate the effects of LBP on steatosis (Supplementary Fig. 2a, b). LBP$^{KI/KI}$ mice exhibited severe steatosis with marked macroscopic changes and aberrant accumulation of LDs both on chow and high-fat diet (HFD) (Fig. 2c, Supplementary Fig. 2c, d). In vitro, primary hepatocytes were isolated and treated with Bodipy C12. LDs in LBP$^{KI/KI}$ hepatocytes were significantly larger compared to those in WT, while negligible LD formation was observed in LBP$^{-/-}$ hepatocytes (Fig. 2d). Further investigation revealed that LBP promotes lipid synthesis in the liver by regulating the AMPK-ACC signaling pathway (Supplementary Fig. 2e–g). Moreover, LBP activates JNK and NF-κB signaling pathways, significantly elevating the expression of inflammation-related molecules such as IL6, ICAM1, and VCAM1. (Supplementary Fig. 2h–k). Moderate liver injury was observed in both the LBP$^{KI/KI}$ and wild-type (WT) groups, as evidenced by similar serum levels of liver enzymes and histological evaluations (Supplementary Fig. 2l–s). We found that the very low-density lipoprotein (VLDL) levels were significantly elevated in LBP$^{KI/KI}$ mice compared to

both the WT and LBP$^{-/-}$ groups, indicating the role of LBP in promoting hepatic TG accumulation (Supplementary Fig. 2t). Consistently, we confirmed that the TG levels undergo similar changes upon overexpressing or knocking down LBP in HepG2 cells (Supplementary Fig. 1d, e). The reintroduction of LBP-GFP to LBP$^{-/-}$ mice resulted in a significant increase in liver TG level, accompanied by the formation of LBP-contained LDs, which was also observed in HepG2 cells and confirmed the role of LBP in promoting LD formation (Fig. 2e, f, Supplementary Fig. 1f). To generate a more detailed visualization of LBP localization in LDs, we introduced LBP-APEX2 into HepG2 cells and conducted transmission electron microscopy (TEM) to observe the ultra-structural assembly of LBP-APEX2 in LDs. Intriguingly, LBP-APEX2 signals were predominantly in larger LDs (Fig. 2g). Taken together, our results strongly suggest that the subcellular localization of LBP within LDs represents a crucial factor in determining of large LDs formation and TG accumulation in hepatocytes.

## LBP promotes long-chain unsaturated triglyceride accumulation in LDs

To investigate the mechanisms underlying steatosis associated with LBP, we conducted lipidomics analysis on liver tissues from LBP$^{KI/KI}$ mice following a 16-week high-fat diet. Our results consistently demonstrated a greater accumulation of triglycerides in LBP$^{KI/KI}$ mice liver, compared to WT mice (Fig. 3a; Supplementary Data 2). Further analysis revealed an upregulation of long-chain polyunsaturated fatty acid-triglycerides (LCPUFA-TG) in LBP$^{KI/KI}$ mice, which are known to be a primary source of lipid peroxidation (Fig. 3b, c). A consistent result was observed in targeted lipidomics analysis, demonstrating that the accumulation of PUFAs and the reduction of short-chain saturated fatty acids (SCSFAs) were detected in the liver tissues of LBP$^{KI/KI}$ mice

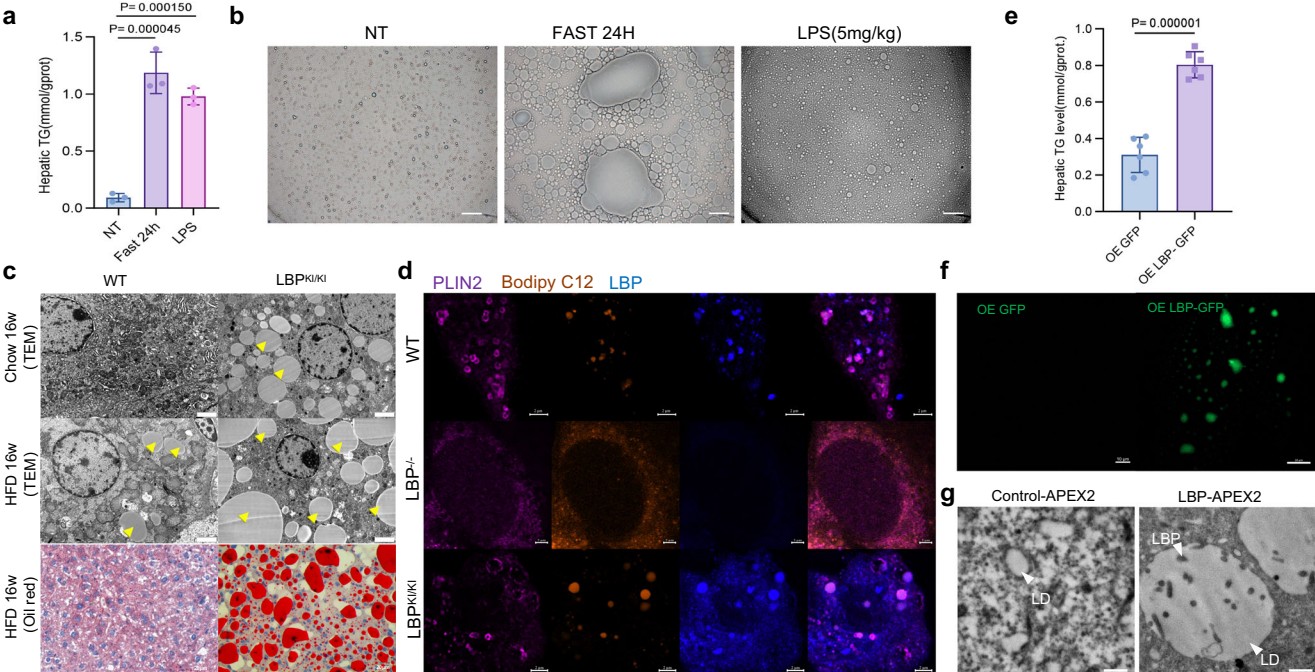

**Fig. 2 | The localization of LBP to LDs drives hepatic steatosis. a** Hepatic TG content of WT mice under NT, 24 h fasting, or LPS-treatment (5 mg/kg, i.p.) for 24 h. Shown are means ± s.d., one-way ANOVA ($n = 3$, biologically independent). **b** Representative bright-field microscope image showing separated LDs from the same treatment as described in (**a**), scale bar = 10 μm. Experiments was conducted with three biological replicates and produced similar results. **c** TEM and Oil red O staining of LDs in the livers of WT and LBP$^{KI/KI}$ mice with 16-week HFD. TEM scale bar = 2 μm, Yellow arrows indicate LDs. Oil red scale bar = 20 μm. Experiments was conducted with three biological replicates and produced similar results. **d** Primary

hepatocytes isolated from WT, LBP$^{-/-}$, LBP$^{KI/KI}$ mice treated with Bodipy C12 (orange) for 16 h and co-stained with PLIN2 (violet) and LBP (blue). Scale bar = 2 μm. Experiments repeated two times independently with similar results. **e** Hepatic TG content of LBP$^{-/-}$ mice overexpressing GFP/LBP-GFP. Shown are means ± s.d., two-tailed unpaired $t$-test ($n = 6$, biologically independent). **f** Representative confocal imaging of LDs separation from liver in (**e**). Scale bar = 10 μm. **g** Representative TEM images of control-APEX2 and LBP-APEX2 morphology, scale bar = 500 nm. Experiments repeated two times independently with similar results.

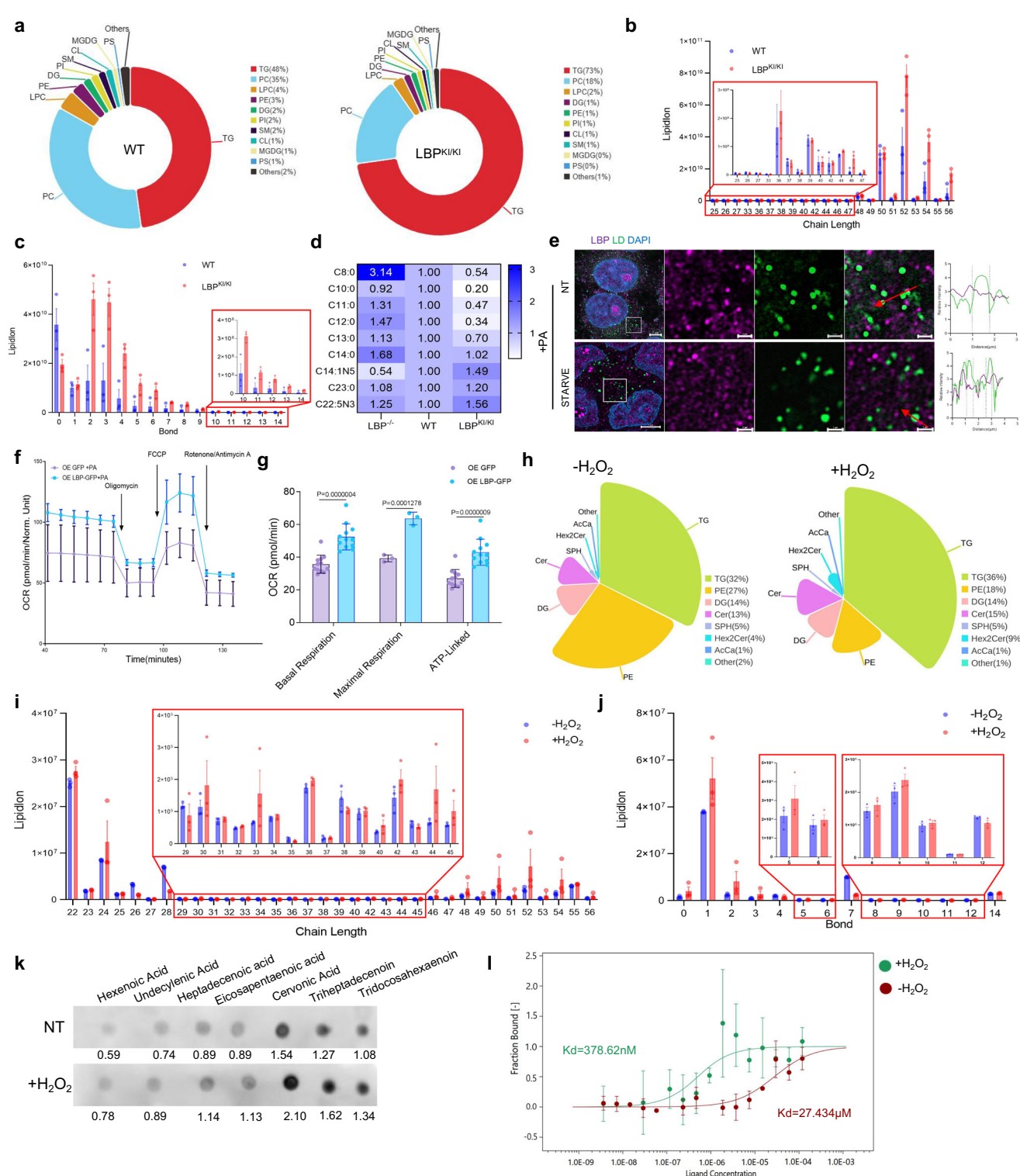

compared to those of WT mice (Fig. 3d). To further confirm the lipidomic data, we added PA to HepG2 cells and found that the LDs formed by the treatment were rarely co-localized with LBP, even under starvation conditions (Fig. 3e). SCSFAs and LCPUFAs are metabolized differently. SCSFAs are more easily decomposed and quickly converted into acyl-CoA in vivo, which then undergoes catabolism through the mitochondrial β-oxidation pathway. In comparison to LCPUFAs, SCSFAs have a shorter metabolic pathway, which require fewer oxygen and electron carriers, lack double bonds, resulting in reduced oxidative stress reactions while rapidly providing energy. Overexpression of LBP enhanced the catabolism of PA in the

mitochondria, leading to an increased demand for mitochondrial function (Supplementary Fig. 3a, b). Using the Seahorse XF96 analyzer, we observed that LBP overexpression significantly increased the oxygen consumption rate (OCR) during PA oxidation, but did not have a similar effect on unsaturated palmitoleic acid (PAO) (Fig. 3f, g, Supplementary Fig. 3c, d). Interestingly, this effect was abolished by the introduction of the F436L mutation (Supplementary Fig. 3e, f), a known loss-of-function mutation of LBP[21]. We further explored the differences in LBP binding to lipids before and after oxidation treatment through immunoprecipitation, and found that following $H_2O_2$ treatment, LBP bound to a greater amount of triglycerides.

**Fig. 3 | LBP binds with unsaturated TG to prevent against lipid peroxidation.**
**a** A pie chart showing the difference in lipid composition between WT ($n = 3$) and
LBP$^{KI/KI}$ ($n = 3$) mice liver after 16 weeks of HFD. **b** Analysis of chain length of dif-
ferential TGs in (**a**). Shown are means ± s.e.m., ($n = 3$, biologically independent).
**c** Analysis of saturation of differential TGs in (**a**). Shown are means ± s.e.m., ($n = 3$,
biologically independent). **d** Targeted quantitative lipidomics illustrating the ratio
of fatty acid content in the liver tissue of LBP$^{KI/KI}$ and LBP$^{-/-}$ groups compared to the
WT group. **e** Confocal microscopy images show co-stained lipid droplets (Bodipy
493/503, green) formed by PA and LBP (violet), with DAPI (blue) counterstaining.
Line graphs represent relative intensity quantification of the area indicated by the
red arrow. Scale bars: 5 μm for the main image, 1 μm for the magnified area.
Experiments were independently repeated three times with consistent results.
**f** Kinetic line graph illustrating the results of Seahorse analyses, indicating that
overexpression of LBP leads to an increase in FAO rate. Shown are means ± s.d.
($n = 4$, biologically independent). **g** Analysis of basal, maximal OCR, and ATP-linked
respiration data from the experiment in (**f**). Shown are means ± s.d., 2way ANOVA
($n = 4$, biologically independent). **h** A pie chart illustrates the changes in lipid
composition bound to LBP before and after 3 h 500 uM H$_2$O$_2$ stimulation. **i** Analysis
of chain length of differential TGs in (**h**). Shown are means ± s.e.m., ($n = 3$, biolo-
gically independent). **j** Analysis of saturation of differential TGs in (**h**). Shown are
means ± s.e.m., ($n = 3$, biologically independent). **k** LBP lipid overlay assay using
purified LBP with non-treatment or 1 h 1 mM H$_2$O$_2$ treatment lipid-bound mem-
branes. **l** Dose-response curves for determination of Kd for the binding of LBP to
tridocosahexaenoin using a labeled MST assay. Shown are means ± s.d., ($n = 3$,
biologically independent).

Additionally, these triglycerides exhibited a preference for long-chain and unsaturated chains (Fig. 3h–j). We selected several fatty acids with varying chain lengths and numbers of unsaturated bonds for an in vitro lipid-binding assay. Our findings revealed that LBP can bind both fatty acids and triglycerides, demonstrating the strongest binding ability towards cervonic acid, which possesses the highest degree of unsaturation and the longest chain among the sample set (Fig. 3k). Notably, following H$_2$O$_2$ treatment, this binding capacity significantly increased. We quantified the interaction dissociation constant (Kd) between LBP and tridocosahexaenoin using microscale thermophoresis (MST). After H$_2$O$_2$ treatment, the Kd value changed from 27.434 μM to 378.62 nM, indicating that oxidation significantly enhanced the LBP-TG binding capacity (Fig. 3l). Based on these findings, we propose a hypothesis that under oxidative stress, LBP facilitates the storage of LCPUFAs in LDs, while promoting the decomposition and energy supply of SCSFAs, suggesting a protective role of LBP in maintaining cellular lipid homeostasis.

## LBP prevents lipolysis and exhibits antioxidative effects

To evaluate the functional consequences of LBP-dependent deposition of LCPUFA-TG, we conducted an integrated analysis of proteomics and transcriptomics (Fig. 4a). The joint analysis revealed that the genes with upregulated expression in the LBP$^{KI/KI}$ group were predominantly associated with TG transport and antioxidant activity, whereas the genes with downregulated expression were primarily related to lipo-lysis (Fig. 4b, c; Supplementary Data 3). To confirm the antioxidant activity of LBP, we observed that LBP$^{KI/KI}$ significantly reduced the levels of protein carbonylation and nitration in the liver of 16-week HFD mice after 24 h fasting, as well as decreasing the level of MDA (Fig. 4d–f). Overexpression of LBP in HepG2 cells significantly decreased the levels of ROS and lipid peroxide resulting from starvation, while promoting the sequestration of peroxidized TG in LDs (Fig. 4g, h). We found that LBP contributes to antioxidant activity by inhibiting lipolysis. LBP effectively suppressed in vitro lipolysis induced by starvation, forskolin (FSK), and isoproterenol (ISO) (Supplementary Fig. 4a–c). Further investigation revealed that LBP reduced the phosphorylated hormone-sensitive lipase (P-HSL) level after ISO treatment (Supplementary Fig. 4d). In vivo lipolysis induced by a 24-h fast was also attenuated by LBP (Supplementary Fig. 4e), as evidenced by reduced translocation of P-HSL to LDs (Supplementary Fig. 4f, g). Consequently, LBP acts as an antioxidant by promoting TG deposition and inhibiting lipolysis to mitigate oxidative damage.

## Antioxidative treatment ameliorates LBP-induced steatosis

To evaluate whether antioxidant therapy could alleviate LBP-induced steatosis, we administered N-acetyl-L-cysteine (NAC), a ROS scavenger, and observed a significant reduction in LD formation and TG accumulation in the livers of LBP$^{KI/KI}$ mice (Fig. 5a, b). The decrease in fasting-induced TG accumulation also demonstrated the beneficial effect of NAC treatment (Supplementary Fig. 5a). NAC treatment not only reduced TG levels but also increased phospholipid levels (Fig. 5c;

Supplementary Data 4), contrary to the trend observed in the lipi-domic data of LBP$^{KI/KI}$ mice liver. Metabolite set enrichment analysis (MSEA) revealed that NAC treatment enhanced phospholipid synthesis (Fig. 5d). We also observed that NAC treatment led to the removal of LBP from LDs (Fig. 5e). To further investigate the effect of phospholipid on the LD homeostasis regulation of LBP, we treated HepG2 cells with NAC or polyene phosphatidylcholine (PC) and observed a significant decrease in the number of LDs and LBP exiting LDs in both treatments. However, we noticed that the efficiency of LBP leaving LDs following PC treatment was lower compared to NAC treatment (Supplementary Fig. 5b). LBP is reported to bind with phospholipid[22], which is produced by ER with common precursors such as TG. We employed Bodipy FL C12-HPC (FL-HPC), a fluorescently labeled phosphati-dylcholine, to chase the interaction between LBP and phospholipids. PC treatment resulted in the exportation of LBP from LDs and its transportation back to the ER, revealing that PC contributes to the shuttle-out process of LBP (Supplementary Fig. 5c). Of note, the pro-portion of FL-HPC to Bodipy C12 affects the efficiency of LBP entering LD (Fig. 5f). We found that FL-HPC was toxic and caused the death of HepG2 cells with LBP overexpression and H$_2$O$_2$ stimulation (Supplementary Fig. 5d, e). These findings led us to hypothesize that a cha-perone may assist LBP in sensing oxidative stress. We searched for proteins interacting with LBP in the IntAct database (www.ebi.ac.uk/intact/home) and identified PRDX4, an ER-localized peroxidase, as a potential candidate with a solid reliability of binding to LBP (Supplementary Fig. 5f). To explore how LBP interacts with PRDX4, we simu-lated the binding mode of LBP and PRDX4 using molecular docking, which revealed that the N-segment of LBP had a high binding affinity for PRDX4 (Supplementary Fig. 5g). We observed that starvation upregulated and induced the interaction of endogenous LBP and PRDX4 in HepG2 cells, which co-localized with LDs (Fig. 5g). Silencing of PRDX4 using small shRNA significantly prevented the export of LBP and promoted LDs growth, while knockdown of LBP eliminated the co-location of PRDX4 and LDs (Fig. 5h). Our results demonstrated a dis-tinct modulation of LBP-induced lipid accumulation and lipolysis inhibition in HepG2 cells by manipulating PRDX4 levels (Fig. 5i, j). Interestingly, the LBP-PRDX4 interaction was abolished by intense oxidative stress, such as H$_2$O$_2$ treatment (Supplementary Fig. 5h, i). Previous studies on PRDX4 suggest that its redox state modulates the conformational transition between the fully folded (FF) and locally unfolded (LU) states[23], which may impact its interaction with LBP. To validate this hypothesis, we utilized a deletion PRDX4$_{1-243}$ and a PRDX4$_{C245A}$ mutation to simulate the LU form and prevent the resolution of oxidized PRDX4. We found that both PRDX4$_{1-243}$ and PRDX4$_{C245A}$ mutants displayed increased binding affinity for LBP N-segment (Supplementary Fig. 5j). This supports the view that the Cys-SOH form of PRDX4 is the major binding state for LBP and confirms the crucial role of N-segment in the PRDX4-LBP interaction (Supplemen-tary Fig. 5g). In addition, the levels of PRDX4 can further influence the regulation of LBP on LDs in the presence of PC (Fig. 5k). Silencing PRDX4 significantly decreased the export of LBP from LDs even when

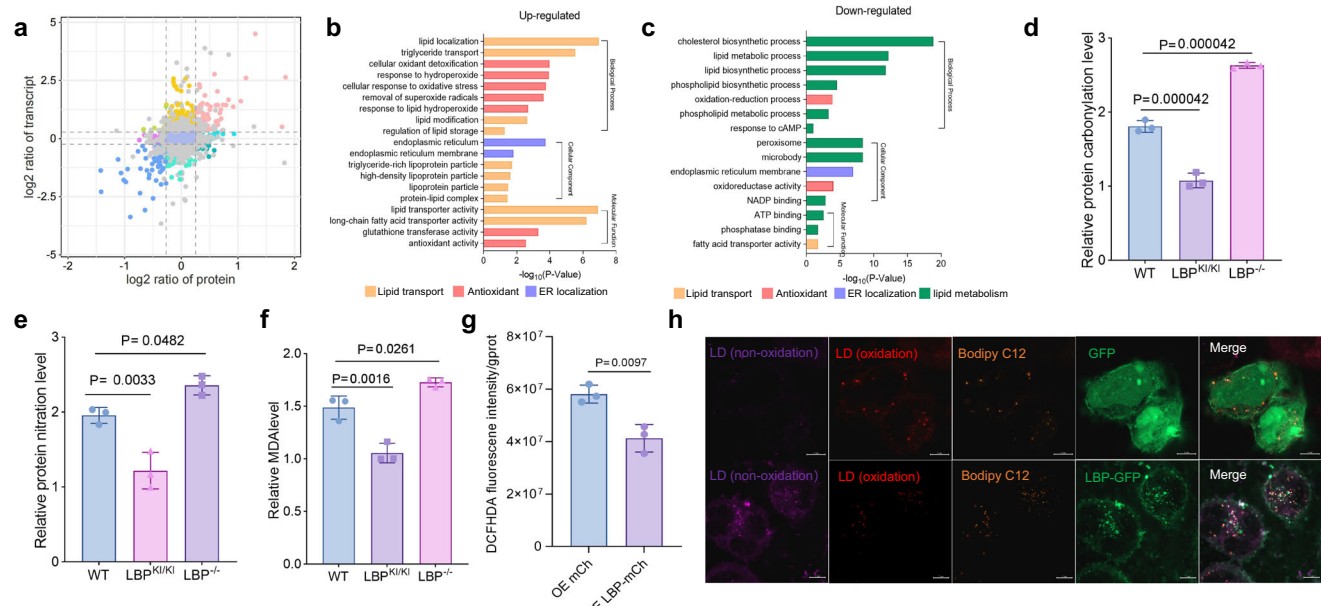

**Fig. 4 | The antioxidant activity of LBP. a** A nine-quadrant plot depicting the combined analysis of the liver transcriptome and proteome of WT and LBP^KI/KI mice fed with HFD for 16 weeks. Each group $n = 3$, biologically independent. **b** Go term enrichment analysis of the differential proteins in the third quadrant (upregulated) of (**a**), using different colors to represent different functions: yellow-lipid transport, red-antioxidant, and purple-ER localization. Bar graph shows the -log10 of $P$-value (hypergeometric test). **c** Go term enrichment analysis of the differential proteins in the seventh quadrant (downregulated) of (**a**), using green color to represent lipid metabolism. Other colors represent the same meanings as in (**b**). Bar graph shows the -log10 of $P$-value (hypergeometric test). **d** Bar graph shows the relative liver protein carbonylation levels of 16-week HFD WT, LBP^KI/KI, and LBP^−/− mice. Shown are means ± s.d., one-way ANOVA, ($n = 3$, biologically independent). **e** Bar graph

shows the relative liver protein 3-nitrotyrosine (3NT) levels of 16-week HFD WT, LBP^KI/KI, and LBP^−/− mice. Shown are means ± s.d., one-way ANOVA, ($n = 3$, biologically independent). **f** Bar graph shows the relative liver malondialdehyde (MDA) levels of 16-week HFD WT, LBP^KI/KI, and LBP^−/− mice. Shown are means ± s.d., one-way ANOVA, ($n = 3$, biologically independent). **g** The bar graph depicts the level of DCFH-DA detected in HepG2 cells following 3 h of nutrient starvation after over-expression of mCherry or LBP-mCherry. Shown are means ± s.d., two-tailed unpaired $t$-test, ($n = 3$, biologically independent). **h** Confocal data showing the co-localization of non-oxidized (violet), oxidized (red) LDs, Bodipy C12 (orange), and GFP/LBP-GFP (green) in 3-h-starved HepG2 cells. Scale bar = 5 μm. Experiments repeated three times independently with similar results.

PC was present while overexpressing PRDX4 reduced LD formation and promoted LBP export in the presence of PC (Supplementary Fig. 5k). Therefore, NAC may affect the LBP-mediated LD homeostasis by modulating the balance of TG and phospholipid, as well as the redox state of PRDX4. These results suggest that the translocation of LBP to LD is modulated by the production balance between TG and phospholipid, which is controlled by the redox state of PRDX4. Therefore, we conclude that LBP couples lipid metabolism with redox signaling, which may largely depend on the lipid-capture capacity of LBP.

## LBP regulates LD homeostasis via the C-segment hydrophobic area and #4-helix TG capture activity

The structure of mouse LBP has been characterized, revealing hydrophobic grooves in C-segments that exhibit greater flexibility. To explore the molecular basis of the lipid-binding capacity of LBP, we employed the AlphaFold AI system (https://deepmind.com/) to predict the 3D structure of LBP, which revealed that the α-helix formed by residues 286-297 (#4-helix) was not observable in the resolved structures present in the PDB database (Fig. 6a). Further secondary structure analysis (http://bioinf.cs.ucl.ac.uk/psipred/) corroborated the prediction of an α-helix (Fig. 6b). Additionally, we discovered that this α-helix is highly conserved among mammals (Fig. 6b). Through de novo modeling of LBP, we obtained results supporting the existence of an α-helix. By hydroxylating the double bond closest to the carboxyl group of small molecule 11094-59-0 (Tridocosahexaenoin) to simulate its oxidized state, we conducted molecular docking using the de novo modeled LBP. The oxidized lipid molecule showed an increased number of hydrogen bonds at the C-segment #4-helix region and exhibited lower binding energy as compared to the prototype

Tridocosahexaenoin (Fig. 6c, d). To further confirm the binding of LBP C-segments to lipids, we observed a co-localization of LBP C-segments (LBP_{SP+216-481}-GFP) with LDs, which was rapidly disrupted by PC (Fig. 6e). We have further identified His_294 on the #4-helix as a potential crucial interaction site between LBP and lipids. The H294A and H294G mutations substantially diminish its ability to translocation to LDs (Fig. 6f). In previous studies, the LBP SNP F436L mutation was found to decrease the cleft of the hydrophobic groove[21]. The molecular simulation model showed that F436L caused ASP-223 to form hydrogen bonds with LYS-443 and TYR-293, which connected the #3 and #4 helices and reduced the cleft in the groove, thereby impeding TG entry LDs (Fig. 6g, h). Consequently, reintroduction of LBP-F436L in LBP^−/− mice resulted in lower hepatic TG level compared to WT LBP (Fig. 6i). In addition, the co-localization of LBP and LDs was inhibited when we disrupted hydrophobic forces using 1,6-hexanediol (Supplementary Fig. 6a). We observed that LBP puncta exhibited liquid-like characteristics and could be dynamically mobilized (Supplementary Fig. 6b; Supplementary Movie 1). Fluorescence recovery after photobleaching (FRAP) experiments showed stronger mobility when binding with TG than PC, suggesting a potential mechanism for the competitive behavior between LBP-PC and LBP-TG (Supplementary Fig. 6c–e). Based on these findings, we propose that the #4-helix and C-segment groove of LBP are critical for capturing and depositing TG. Collectively, our findings suggest that LBP levels may increase in response to oxidative stress. Both endogenous and exogenous LBP C-terminus capture TG and bind PRDX4 via its N-terminus, entering LDs to play an antioxidant role. When oxidative stress dissipates, PRDX4 senses the oxidation signal and switches from the oxidized to the reduced state, assisting LBP leaving LDs, thereby promoting LDs lipolysis (Fig. 6j).

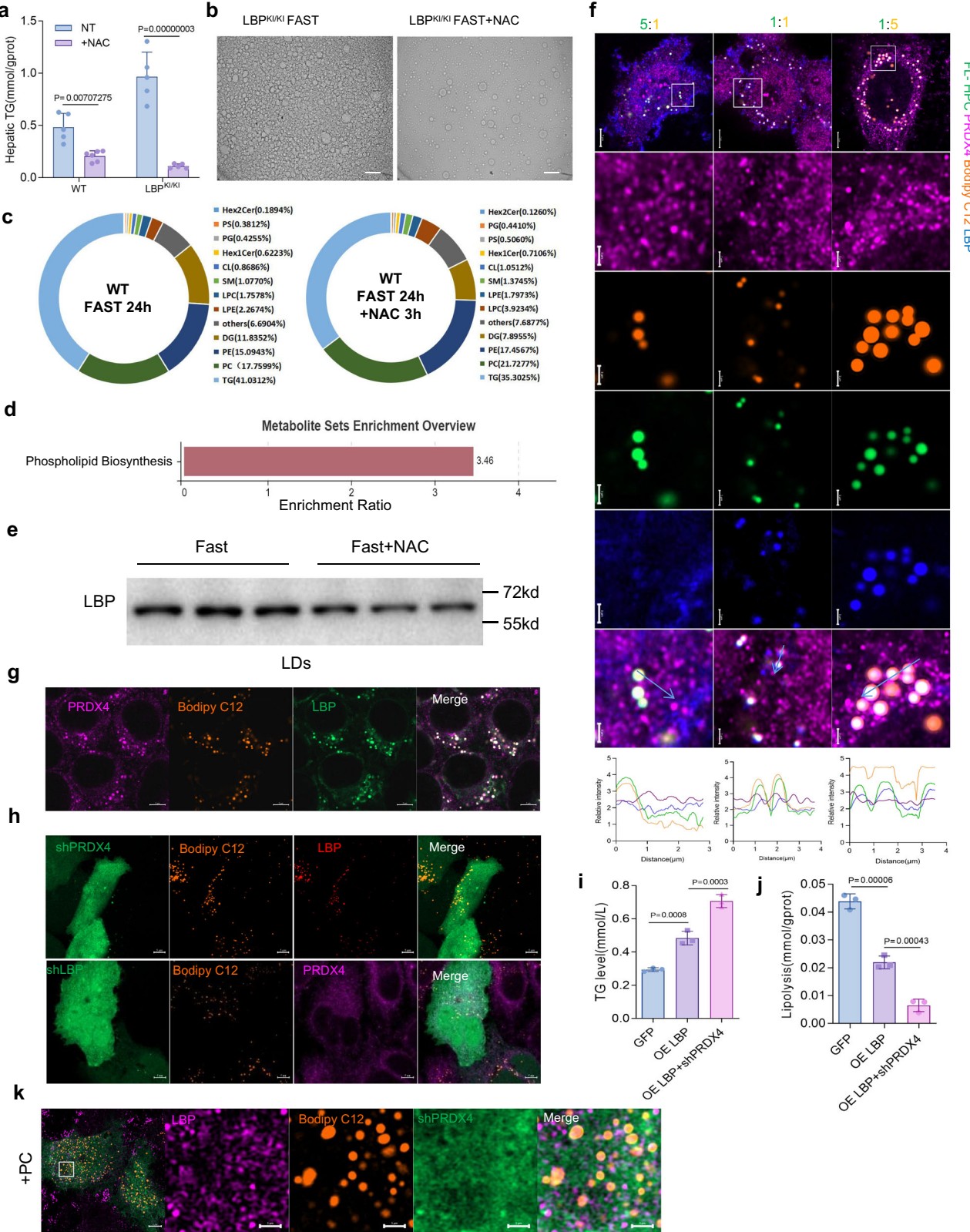

## Chronic high expression of LBP induces obesity

As a known regulator of metabolism, continuous upregulation of LBP results in severe steatosis. Nevertheless, it remains unclear whether LBP is involved in responding to chronic stress and contributing to systemic dysregulation. We investigated the systemic effects of LBP in the context of fatty liver. Hepatic lipid export was assessed using the non-ionic detergent poloxamer 407, which inhibits lipoprotein lipase activity. We observed significantly higher systemic TG levels in LBP$^{KI/KI}$ mice compared to WT and LBP$^{-/-}$ mice, suggesting an increased hepatic TG export (Fig. 7a). After 16 weeks of HFD, LBP$^{KI/KI}$ mice developed obesity and an increased accumulation of adipocytes (Fig. 7b). Additionally, 16-week HFD LBP$^{KI/KI}$ mice had higher fasting blood glucose and insulin levels (Fig. 7c, d). The glucose tolerance test (GTT) and insulin tolerance test (ITT) revealed that knockin of

**Fig. 5 | Redox signaling regulates LBP-induced LD accumulation via phospholipid biogenesis and the LBP chaperone function of PRDX4. a** The bar graph compares the hepatic TG levels in WT mice and LBP[KI/KI] mice, with/without NAC treatment. Means ± s.d., 2way ANOVA ($n = 5$–6, biologically independent). **b** Brightfield microscope image of LDs in LBP[KI/KI] mice, fasted for 24 h and treated with/without NAC treatment for 3 h. Scale bar = 10 μm Experiments repeated three times independently with similar results. **c** A pie chart representing the difference in lipid composition between WT mice that were fasted for 24 h ($n = 3$) and those treated with NAC for 3 h ($n = 3$). **d** MSEA analysis for the lipidomic data in (**c**). **e** Changes in LBP content within LDs extracted from liver tissue with/without NAC treatment for 3 h in 24 h fasting mice ($n = 3$, biologically independent). **f** Confocal microscopy images (upper panel) and quantification (lower panel) of LBP (blue) co-stained with PRDX4 (violet) at different ratios of Bodipy C12 (orange) and FL-HPC (green). Scale bar = 5 μm, magnified area scale bar = 1 μm. Experiments repeated three times independently with similar results. **g** Staining of LBP (green) and PRDX4 (violet) co-localized with Bodipy C12 (orange) in HepG2 cells after 16 h of Bodipy C12 treatment and 3 h of starvation. Scale bar = 5 μm. Experiments repeated two times independently with similar results. **h** Confocal images show that knockdown of PRDX4 does not affect the localization of LBP in LD, but increases LD size, whereas knockdown of LBP affects PRDX4 localization in LD. The shRNA scaffold plasmid with dual promoters for expressing green fluorescent protein; red represents LBP; violet denotes PRDX4, while Bodipy C12 is orange. Scale bar = 5 μm. Experiments repeated three times independently with similar results. **i** Effects of PRDX4-knockdown on LBP-mediated regulation of TG accumulation. Means ± s.d., one-way ANOVA ($n = 3$, biologically independent). **j** Effects of PRDX4-knockdown on LBP-mediated regulation of lipolysis. Means ± s.d., one-way ANOVA ($n = 3$, biologically independent). **k** Representative confocal image shows that shPRDX4 (green) reduces the ability of LBP (violet) to leave the LD (Bodipy C12, orange) after 3 h of PC treatment. Scale bar = 5 μm, magnified area scale bar =1 μm. Experiments repeated three times independently with similar results.

LBP-induced significant insulin resistance and glucose tolerance phenotypes (Fig. 7e–h). We further investigated the effect of LBP on glucose metabolism. PAS staining revealed that LBP[KI/KI] mice had reduced glycogen accumulation in their livers (Fig. 7i). Seahorse assay results showed that inhibiting amino acid metabolism with BPTES did not significantly change metabolic status with LBP overexpression, suggesting cells primarily do not use amino acids as an energy source (Fig. 7j, k). Inhibiting glucose metabolism with UK5099 enhanced OCR with LBP overexpression, indicating a preference for lipid catabolism to supply energy (Fig. 7l, m). Inhibiting fatty acid metabolism with Etomoxir led to a significant decrease of OCR with LBP overexpression, indicating less efficient glucose utilization (Fig. 7n, o). We assessed the respiratory exchange ratio (RER) of WT, LBP[KI/KI], and LBP[−/−] mice using a metabolic cage assay (Fig. 7p). A ratio closer to 1 indicates predominantly glucose metabolism, whereas a ratio closer to 0.7 suggests reliance on fatty acid metabolism[24]. The results demonstrated that LBP[KI/KI] mice were more inclined to use fatty acid metabolism as an energy source compared to WT and LBP[−/−] mice. These results indicate that elevated hepatic LBP expression is associated with systemic adipocyte accumulation, obesity, and insulin resistance.

Consistent with a recent report[25], chronic stress induces an increase in LBP expression. We observed a significant elevation of serum LBP levels after four weeks of consecutive chronic jet lag (CJL) or forced swimming test (FST), which was accompanied by increased hepatic TG, serum TG, and body fat ratio (BFR) (Supplementary Fig. 7a–d). Interestingly, liver LD exhibited elevated LBP levels and diminished PRDX4 levels (Supplementary Fig. 7e). The body weight of mice in both chronic stress groups significantly increased compared to the control group, while food intake decreased (Supplementary Fig. 7f, g). Chronic stress mice had higher fasting insulin levels and exhibited insulin resistance and glucose tolerance phenotypes (Supplementary Fig. 7h–l). These results suggest that chronic stress may upregulate LBP levels, which can mediate TG accumulation and result in obesity under stress.

Next, we aimed to determine how to reverse LBP-induced TG accumulation. NAC treatment, but not starvation, effectively induced lipolysis of peripheral adipocytes in LBP[KI/KI] mice with HFD (Fig. 7q–s). NAC treatment for 3 h reversed the starvation-induced TG accumulation in LBP[KI/KI] mice and was associated with increased serum free fatty acid (FFA) levels and decreased TG levels (Fig. 7t, u). Interestingly, feeding the mice a ketogenic diet with 90% saturated fat did not cause fatty liver or obesity in any of the LBP[KI/KI], WT, or LBP[−/−] groups (Fig. 7v), which was consistent with our previous finding that LBP binds with PUFA-TG. In conclusion, our findings suggest that increased LBP expression results in not only fatty liver but also peripheral fat accumulation that leads to obesity. Antioxidant therapy shows promise as a treatment for obesity caused by LBP.

## Discussion

In this study, through a combination of RNA-seq and proteinomics detection method we characterized LBP within LDs under oxidative stress. Previous studies have elucidated several pathways underlying the observed augmentation in LDs induced by ROS exposure, consistent with our sequencing data[26]. However, our findings diverge from the conventional paradigm by delineating LBP as an effector protein to regulate LD homeostasis under cellular oxidative stress. We identified functions and implications of LBP in cellular processes, including its role as an antioxidant, mediator of lipid metabolism, and participating in redox signaling pathways.

Metabolic Associated Fatty Liver Disease (MAFLD), as a manifestation of metabolic syndrome in the liver[27], currently lacks effective therapeutic strategy[28]. LBP, as an obesity-related insulin resistance biomarker[29], is closely related to MAFLD, which is confirmed by our study. Prior studies have demonstrated that circulating LBP levels are significantly elevated in MAFLD, and the level of LBP play a regulatory role in high-fat and high-sucrose diet-induced MAFLD, with knockdown or knockout of LBP both significantly reducing lipid accumulation in liver[16,17]. However, the current limitations of the studies lie in its focus on the phenotypic observation of the therapeutic effect of LBP knockout on MAFLD. Considering the high expression of LBP in the liver and its crucial role in inflammatory responses, knocking out LBP may not be an ideal intervention strategy. Through the knockin of LBP, we obtained consistent animal phenotypes with the previous study, demonstrating that the binding of LBP to lipids entering LDs directly leads to the development of fatty liver. It is notable that we discovered LBP[KI/KI] mice were unable to induce obesity under a ketogenic diet. Our ketogenic diet feed consists of 90% saturated fatty acids, which aligns with the conclusion presented in this study that LBP protects long-chain unsaturated TGs from entering LDs rather than saturated TG. In addition, the ketogenic diet alters the energy source, transforming FFA into ketone bodies for fuel. This dietary method has been shown to decrease peripheral insulin levels and augment mitochondrial respiration[30]. It also inhibits the activation of the AMPK pathway[31], thus hindering lipid accumulation. Furthermore, a ketogenic diet suppresses oxidative stress[32]. These combined factors help to prevent the onset of obesity caused by LBP.

Our findings suggest that the expression of LBP is susceptible to regulation by ROS signaling. Oxidative stress is a major factor in MAFLD, playing a crucial role in liver steatosis[33]. Previous studies have shown decreasing the expression level of LBP significantly enhances the oxidative stress in the liver under both physiological and pathological conditions[14,15]. Our data fill the gap in the understanding of the antioxidative role of LBP and suggest that LBP is involved in MAFLD. In this study, we demonstrate that the promotion of lipid accumulation by LBP may be due to its antioxidant protective function. If the oxidative stress is not eliminated, using phosphatidylcholine to promote LD decomposition could lead to severe damage. Polyene

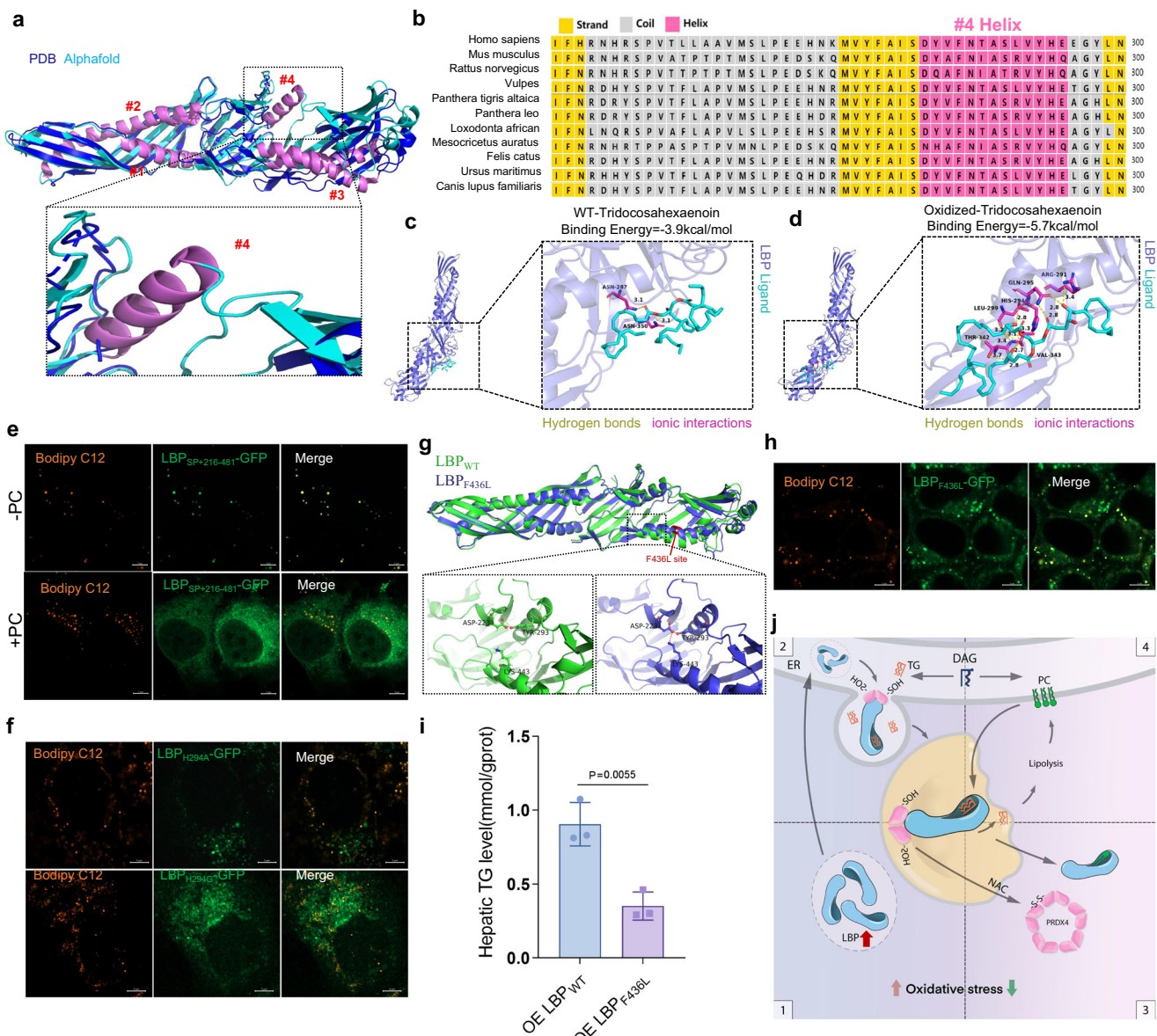

**Fig. 6 | The #4-helix within the C-segment of LBP interacts with oxidized TGs.**
**a** LBP structure solved in the PDB database and predicted by Alphafold, with the black box indicating the differential #4α-helix. **b** Predict secondary structures and perform homology comparisons of LBP in various mammals. **c** Molecular docking diagram of WT-Tridocosahexaenoin with LBP. LBP is represented as a dark blue cartoon model, while the ligand is depicted as a cyan stick model. The binding site is shown as a magenta stick structure. (yellow: Hydrogen bonds, magenta: ionic interactions, and green dashed lines: hydrophobic interactions). **d** Molecular docking diagram of oxidized-Tridocosahexaenoin with LBP. **e** Representative co-stained images captured by confocal microscope showing the expression of C-segment LBP-GFP (LBP_SP+216-481-GFP, green) for 48 h, treated with Bodipy C12 (orange) or Bodipy C12+PC for 16 h. Experiments repeated three times independently with similar results. **f** Representative co-stained images captured by confocal microscope showing the expression of H294A mutation (green) and H294G

mutation (green) for 48 h, treated with Bodipy C12 (orange) for 16 h. Experiments repeated five times independently with similar results. **g** Comparison of Alphafold predicted LBP_WT(green) and LBP_F436L(blue) protein structures. The F436L site is indicated in red (upper panel). **h** Representative confocal images of expression LBP_F436L-GFP (green) for 48 h and co-stained after adding Bodipy C12 (orange) for 16 h. Experiments repeated five times independently with similar results. **i** Hepatic TG content of LBP$^{-/-}$ mice with overexpressing LBP_WT-GFP or LBP_F436L-GFP. Shown are means ± s.d., two-tailed unpaired t-test (n = 3, biologically independent). **j** Working model for LBP and PRDX4 interactions: 1. LBP expression upregulated response to oxidative stress; 2. LBP translocation to LD for promoting LD growth; 3. PRDX4 resolution promoting the export of LBP from the LDs; 4. Export of LBP enhancing lipolysis and phospholipid synthesis. Illustration by Mo Liu, with permission.

phosphatidylcholine is a common drug for clinical treatment of fatty liver[34]. Previous studies have determined that stress leads to adverse metabolic effects by elevating the formation of ROS[35,36]. Chronic stress causes upregulation of LBP, which functions to buffer ROS overload, potentially serving as a mechanism underlying stress-induced obesity. This effect prevents tissue injury and systemic dysfunction, which may represent metabolically healthy obesity[37]. Our study provides a reference for its therapeutic scenarios. The use of Polyene

phosphatidylcholine may result in greater liver damage when the fatty liver is caused by acute starvation or other acute oxidative stress.

Although the core of LDs is widely considered to be a hydrophobic environment primarily composed of neutral lipids, there have been reported of polar molecules, including certain cytoplasmic proteins, such as PAT family proteins and cyclooxygenases, inside LDs[38,39]. However, the exact mechanism underlying the incorporation and localization of these polar molecules within LDs remain to be fully

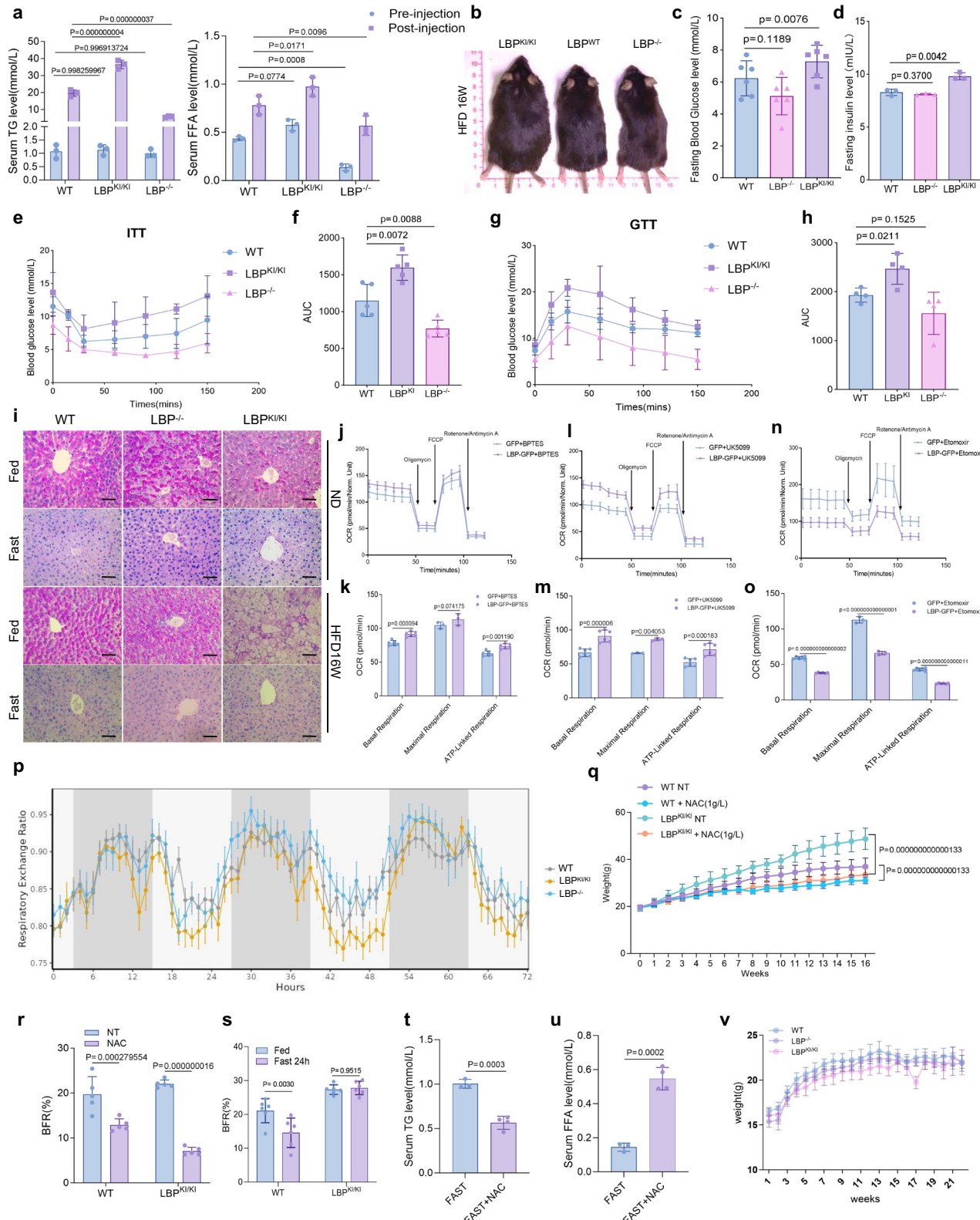

elucidated. Our study provides insight into the ability of LBP to enter LDs, which we attribute to its C-segment hydrophobic groove lipid-binding capacity. Our docking result revealed the lipid-binding pocket of LBP formed more hydrogen bonds with oxidized triglycerides, indicating the antioxidant capacity of LBP. The lipidomics analysis of LBP$^{KI/KI}$ liver and LBP immunoparticipation were in accordance with this result and showed that LCPUFA-TG, which is more prone to be oxidized, was preferentially sequestered within the LDs. We found that

the #4-helix within the C-segment of LBP is undetectable in the x-ray crystal structure. We hypothesized the existence of an α-helix through secondary structure prediction, homology modeling, and de novo modeling approaches, and confirmed the importance of the hydrogen bond formed by His_294 of the #4-helix with lipids for LBP lipid-binding capacity via molecular docking and molecular dynamics simulations. Previous studies may have missed a stable α-helix due to the absence of lipid ligands, leading to structural instability.

**Fig. 7 | LBP upregulation drives obesity and is rescued by antioxidant treatment. a** Serum TG and FFA levels in 8-week-old mice before and after Poloxamer 407 injection. Means ± s.d., 2way ANOVA ($n = 3$, biologically independent). **b** Morphological comparison of LBP$^{KI/KI}$, LBP$^{-/-}$ and WT mice fed on a 16-week-fat diet. **c** Fasting blood glucose levels of mice in (**b**). Means ± s.d., two-tailed unpaired $t$-test ($n = 6$, biologically independent). **d** Fasting insulin levels of mice in (**b**). Means ± s.d., two-tailed unpaired $t$-test ($n = 3$, biologically independent). **e** Serum glucose levels during insulin tolerance tests in mice on a 22-week high-fat diet. Means ± s.d. ($n = 5$, biologically independent). **f** Area under curve (AUC) analysis of (**e**). Means ± s.d., two-tailed unpaired $t$-test ($n = 5$, biologically independent). **g** Serum glucose levels during glucose tolerance tests in mice on a 22-week high-fat diet. Means ± s.d. ($n = 4$ biologically independent). **h** AUC analysis of (**g**). Means ± s.d., two-tailed unpaired $t$-test ($n = 4$, biologically independent). **i** PAS staining of mice liver sections, scale bar = 20 μm. Three independent experiments with similar results. **j** Mitochondrial respiration rates in control/LBP-overexpressing cells with BPTES. Means ± s.d. ($n = 3$, biologically independent). **k** Statistical analysis of (**j**). Means ± s.d., 2way ANOVA ($n = 3$, biologically independent). **l** Mitochondrial respiration rates in control/LBP-overexpressing cells with UK5099. Shown are means ± s.d. ($n = 3$, biologically independent). **m** Statistical analysis of (**l**). Means ± s.d., 2way ANOVA ($n = 3$, biologically independent). **n** Mitochondrial respiration rates in control/LBP-overexpressing cells with Etomoxir. Shown are means ± s.d. ($n = 3$, biologically independent). **o** Statistical analysis of (**n**). Means ± s.d., 2way ANOVA ($n = 3$, biologically independent). **p** Respiratory exchange ratio of mice in a metabolic cage. Means ± s.d. ($n = 5$, biologically independent). **q** Body weight curves of mice with/without NAC treatment. Means ± s.d., 2way ANOVA ($n = 5–6$, biologically independent). **r** Body fat ratio of mice with/without NAC treatment. Means ± s.d., 2way ANOVA ($n = 5$, biologically independent). **s** Body fat ratio of mice with/without 24 h fasting. Means ± s.d., 2way ANOVA ($n = 6$, biologically independent). **t** Serum TG levels in 24 h fasted LBP$^{KI/KI}$ mice with/without NAC recovery. Means ± s.d., two-tailed unpaired $t$-test ($n = 3–4$, biologically independent). **u** FFA levels in 24 h fasted LBP$^{KI/KI}$ mice with/without NAC recovery. Means ± s.d., two-tailed unpaired $t$-test ($n = 3–4$, biologically independent). **v** Body weight curves of mice under a ketogenic diet. Means ± s.d., 2way ANOVA ($n = 6$, biologically independent).

Furthermore, we observed that the ability of the F436L mutant to enter LDs was diminished, which was consistent with previous research that found this mutation to be associated with decreased BMI[40].

Our research also yielded interesting findings concerning the molecular chaperone PRDX4 of LBP. PRDX4 is a well-recognized antioxidant protein, which has been documented for its protective effects against hepatic steatosis[41]. Previous studies have established that LBP and PRDX4 interact with each other. Given their common endoplasmic reticulum localization, this feature renders them capable of synergistically modulating LD biogenesis. Our investigations further reveal that PRDX4, serving as the oxidative sensor for LBP, plays a crucial role in the shuttling of LBP within LDs. Importantly, these results may suggest that LBP can protect peroxidized PRDX (PRDX4-SOH) by secluding PRDX4 into LDs. This not only prevents PRDX4 from hyper-oxidation(PRDX4-SO$_2$H) but also facilitates its restoration to the PRDX4 reaction cycle's resolution step, ensuring the proper functioning of this vital antioxidant protein. The metabolic balance between the synthesis of TG and phospholipid forms a complementary mechanism of cell redox damage prevention[42,43]. PC, which has the ability to emulsify fat, is a classic anti-fatty liver drug[44]. However, our research has discovered that in the absence of elimination of oxidative stress, administration of PC may result in the loss of the protective effect of LBP on LDs and cause more cellular damage. Of interest, NAC treatment resulted in an increase in cellular phospholipid levels. Additionally, NAC promoted the resolution and decamerization of PRDX4, which facilitated the substitution of LBP/TG with LBP/PC and activated lipolysis after the elimination of ROS (Fig. 6j).

The current study has a number of strengths. Based on the known buffer effect of LDs on ROS, we identified the LBP plays a key regulatory role in this process. Through multi-omics, biochemical methods, and various mutant analyses, it has been determined that LBP is located in LDs and plays an important role in antioxidant processes. By observing the phenotype in LBP$^{-/-}$ and LBP$^{KI/KI}$ mice, as well as through multi-omics studies, the protective effects of LBP on UFA-TG have been established. Furthermore, PRDX4 has been identified as an oxidative sensor of LBP. It has also been clarified that redox signals regulate the synthesis of TG and PC, thereby influencing the transportation activity of LBP within LD. This study provides a therapeutic target for fatty liver induced by LBP and suggests that anti-oxidative treatment strategies may be more effective than the use of phosphatidylcholine alone. However, further investigation is required to fully understand the mechanism by which LBP modulates LDs and to which extent it contributes to LBP-mediated obesity. Examining the biophysical characteristics of LDs localized with LBP could provide valuable insights. Additionally, it remains unclear whether chronic upregulation of LBP contributes to the development of stress-related metabolic disorders, such as diabetes and cardiovascular disease.

Subsequent in-depth studies are necessary for deeper comprehension. Moreover, a correlational examination between LBP and stress levels via a population-based questionnaire survey could offer a significant value.

In conclusion, we uncovered the mechanism underlying oxidative stress-induced accumulation of LDs. LBP act as an antioxidant, interfacing with the lipid metabolism and redox signaling pathways, precisely maintains cellular homeostasis to adapt to the oxidative stress. Controlling the LBP pathway under oxidative stress shows promise for targeted interventions in stress-induced obesity and metabolic dysfunction, opening avenues for refined redox medicine in the future.

# Methods
## Materials
LBP Polyclonal antibody (WB:1:1000, IF:1:200,Cat.#23559-1-AP, Lot.#00017794), LBP Monoclonal antibody (WB:1:1000, IF:1:200, Cat.#66181-1-Ig, Lot.#10001902), Calnexin Polyclonal antibody (WB:1:1000, IF:1:200, Cat.#10427-2-AP, Lot.#00093709), HSL Polyclonal antibody (WB:1:2000, IF:1:200, Cat.#17333-1-AP, Lot.#00055498), ADRP/Perilipin 2 Polyclonal antibody (WB:1:2000, IF:1:200, Cat.#15294-1-AP, Lot.#00093566), PRDX4 Polyclonal antibody(WB:1:2000, IF:1:200, Cat.#10703-1-AP, Lot.#00051410), TOM20 Monoclonal antibody (WB:1:2000, IF:1:200, Cat.#66777-1-Ig, Lot.#10020635), CoraLite488-conjugated Goat Anti-Mouse IgG(H+L) (1:500, Cat.#SA00013-1, Lot.#20000422), CoraLite594-conjugated Goat Anti-Rabbit IgG(H+L) (1:500, Cat.#SA00013-4, Lot.#20000239), HRP-conjugated Affinipure Goat Anti-Rabbit IgG(H+L) (1:3000, Cat.#SA00001-2, Lot.#20000798) and HRP-conjugated Affinipure Goat Anti-Mouse IgG(H+L) (1:3000, Cat.#SA00001-1, Lot.#20000325) were purchased from Proteintech Inc.; Phospho-HSL (Ser660) Antibody (WB:1:2000, IF:1:200, Cat.#AF8026, Lot.#40u3105) was from Affinity Biosciences Inc.; Anti-Catalase Mouse mAb (WB:1:2000, IF:1:200, Cat.#PTM-5630, Lot.#ML070144), Anti-Caveolin-1 Rabbit mAb (WB:1:2000, Cat.#PTM-5056, Lot.#L030327) were from PTM Bio Inc.; β-actin Mouse Monoclonal Antibody (WB:1:5000,Cat.#E12-041, Lot.#EG20200316) and mCherry-Tag mouse monoclonal antibody (WB:1:2000,Cat.#E12-010, Lot.#EG20200620) were from EnoGene Biotech Inc.; Goat Anti-Rabbit IgG (H&L)-Alexa Fluor 647(1:500, Cat.#RS3811, Lot.#B1101RA13) was from Immunoway; p-AMPKα(T182/T172) antibody (WB:1:1000, Cat.#WL05103, Lot.#R08215103), AMPKα antibody (WB:1:1000, Cat.#WL02254, Lot.#R08232254), IκBα antibody (WB:1:1000, Cat.#WL01936, Lot.#R08241936), p-IκBα (ser32/ser36) antibody (WB:1:1000, Cat.#WL02495, Lot.#R06132495), JNK antibody (WB:1:1000, Cat.#WL01295, Lot.#L03041295), p-JNK (Thr183/Tyr185) antibody (WB:1:1000, Cat.#WL01813, Lot.#L03161813) and malondialdehyde (MDA) assay kit (WLA048) were from Wanleibio Inc.;

Goat Anti-Mouse IgG H&L / AF350 antibody (1:500, Cat.#bs-0296G-AF350, Lot.#AIO4125842)was from Bioss Inc.; lipopolysaccharides from Escherichia coli O55:B5 (L2880) was purchased from Sigma Inc.; Tunicamycin (HY-A0098) and Poloxamer 407 (HY-D1005) were purchased from MCE Inc.; Atglistatin(T1875) and Forskolin (T2939) were purchased from TargetMol Inc.; N-Acetylcysteine amide (38520-57-9) was purchased from Macklin Inc.;Polyene Phosphatidylcholine(H20057684) was purchased from Chengdu Tiantai Mountain Pharmaceutical Inc.; Isoprenaline (H31021344) was purchased from Shanghai Hefeng Pharmaceutical Inc.; Lipid Droplet Isolation Kit (MET-5011) was purchased from Cell Biolabs Inc.; Membrane Protein Extraction Kit (C500049) was purchased from Sangon Biotech Inc.; 3-Nitrotyrosine Elisa kit (EU2560) and Protein Carbonyl Elisa kit (EU2629) were purchased from FineTest Ltd.; TG (A110-1-1) and FFA (A042-2-1) assay kits were purchased from Nanjing Jiancheng Bioengineering Institute; Bodipy 493/503 (GC42959) was purchased from GLPBIO; Bodipy 665/676 (B3932), Bodipy 558/568 C12 (D3835) and β-Bodipy FL C12-HPC (D3792) were purchased from Invitrogen.

## Mouse experiments

All animal experiments were approved by the Animal Care and Use Committee (2021-N(A)-144) of the First Affiliated Hospital of China University of Science and Technology. WT male mice, LBP$^{-/-}$ male mice, and LBP$^{KI/KI}$ male mice were prepared by GemPharmatech Inc. (Nanjing, China), utilizing the C57/BL 6J strain as the foundation. The following two pairs of forward and reverse primers were used for LBP$^{-/-}$ genotyping: AGTCAGGGACATTTGTCTTCTGCC and TTCTACCCTAAGCAGGGAAGAGC, GCTCTGGTAAGTGTCCAGGATTGG and CAGCATCAGTATGGTTGGCTCCAG. The following three pairs of forward and reverse primers were used for LBP$^{KI/KI}$ genotyping: ATGCCCACCAAAGTCATCAGTGTAG and ATACTGAGTGGATGCTGTAACGCAG, CCTCCTCTCCTGACTACTCCCAGTC and TCACAGAAACCATATGGCGCTCC, CAGCAAAACCTGGCTGTGGATC and ATGAGCCACCATGTGGGTGTC. All animals were housed under standard conditions with a temperature of 23 °C, humidity at 60%, and unlimited access to water. To investigate the role of LBP in the induction of steatosis, the mice were fed with a chow diet (Xietong Bio Inc., Cat.#1010088), 60% HFD (Xietong Bio Inc., Cat.#XTHF60), or 90% ketogenic diet (Research Diets, Inc., Cat.#D10070801) for 16 weeks, with or without 24 h fasting. Moreover, treatment of NAC (Macklin Inc., Cat.#616-91-1, 1 g/L in drinking water) or polyene phosphatidylcholine (10 mg/kg, intragastric) was performed to investigate the antioxidant role of LBP. In the rescue experiment, 5-week-old WT and LBP$^{KI/KI}$ mice were subjected to a 16-week 60% high-fat diet, and NAC treatment (1 g/L in drinking water) was given starting from the beginning of the HFD. In the acute experiment, WT mice were fasted for 24 h, and fed for 3 h, with the treatment of NAC (500 mg/kg, intragastric) or polyene phosphatidylcholine (10 mg/kg, intravenous).

In order to confirm the response of LBP to chronic stress, the forced swimming and chronic jet lag were performed on mice with a chow diet or 60% HFD. The forced swimming test of 15 min per day was carried out as described previously[45]. Chronic jet lag was performed by repeating light and night circles of 8 h/16 h, 8 h/16 h, and 8 h/24 h for 30 days as described in the earlier study[46]. The mice in 12 h/12 h light and dark circles were set as control.

## Cell culture

HepG2 (iCell-h092) and HEK-293T (iCell-h237) cells were obtained from the iCell Bioscience Inc. (Shanghai China), and cultured in high glucose DMEM media (VivaCell Biosciences), which was supplemented with 10% FBS (ExCell Bio) and 1% streptomycin/penicillin (Biosharp) at 37 °C, containing 5% $CO_2$. Primary hepatocytes were derived from different genotypes of 8-week-old mice according to published procedures[47]. The cells were seeded on collagen-coated plates. After overnight preculture, the cells were treated with 1 μM final

concentration Bodipy 558/568 C12 for 24 h. Then the cells were subjected to Immunofluorescence staining.

## Plasmid construction and transient transfections

To Knockdown, Cloning the shRNA sequences targeting human LBP (5'-3': AAGGCCTGAGTCTCAGCATCTC) and PRDX4 (5'-3': ACCTGGTAGTGAAACAATA) into pGreenPuro hairpin lentivector (SBI, Cat.#SI505A-1,Lot.#221017-005) separately. By inserting target sequence into pLVX-Green1-N1 vector, pLVX-Cherry-N1 vector (MiaoLingBio, Cat.#632562, Lot.#ML20221021) or pLVX-BFP-N1 vector (MiaoLingBio, Cat.#P0474, Lot.#ML20190319), the overexpression vectors were generated. All plasmids were constructed by restriction enzyme digestion (Yugong Biolabs) and T4 ligation (Monad Biotech) methods and were confirmed by Sanger sequencing. Nucleic acid concentration was measured using the Onedrop spectrophotometer (WuYi, China) and the OD-2000C software (version V3.7.29). The plasmid was transfected with Lipofectamine 2000 (Invitrogen) according to the manufacturer's instructions. After the indicated transfection times, various functional studies were conducted.

## RNA sequencing

We conducted transcriptomic sequencing on LBP$^{-/-}$, LBP$^{KI/KI}$, and their WT control mice that were fed a high-fat diet for 16 weeks, with three biological replicates per group. Total RNA was extracted from Fresh liquid nitrogen quick-frozen tissue using the Trizol reagent kit (Invitrogen) according to the manufacturer's protocol. The quality of RNA was assessed on an Agilent 2100 Bioanalyzer (Agilent Technologies) and quantified by NanoDrop and Qubit. The mRNA was enriched by Oligo (dT) beads and fragmented into short fragments. The cDNA was reverse transcribed by using NEBNext Ultra RNA Library Prep Kit for Illumina (NEB). The purified double-stranded cDNA fragments were added with one base and ligated to the Illumina sequencing adapter. The ligation was purified with the AMPure XP Beads (1.0X). The size of the ligated fragments was selected by agarose gel electrophoresis and amplified by polymerase chain reaction. The resulting cDNA library was sequenced using Illumina Novaseq 6000 by Gene Denovo Biotechnology Co. (Guangzhou, China). We used open-source software tools Bowtie2/HISAT, String-Tie, and DESeq2 pipeline for analysis of RNA-seq experiments as previously described[48]. We next adopted Metascape (http://metascape.org) for functional annotation of the differentially expressed genes.

## Lipid droplets isolation and 4D proteomic analysis

All proteomics experiments in this study used three independent biological replicates for protein extraction and testing per group. The LDs proteins were identified by the LC−MS/MS analysis, which was performed by PTM BIO Inc.100 mg of liver tissue was weighed, and the lipid droplet isolation experiment was performed using the Lipid Droplet Isolation Kit according to the manufacturer's instructions. The obtained lipid droplets were grinded with liquid nitrogen into powder, and four volumes of lysis buffer (8 M urea, 1% protease inhibitor) were added respectively, followed by sonication three times on ice using a high-intensity ultrasonic processor (Scientz). The remaining debris was removed by centrifugation at 12,000 × g at 4 °C for 10 min. Finally, the supernatant was collected, and the concentration of protein was measured by the BCA kit (Beyotime) according to the manufacturer's instructions. For trypsin digestion, DTT with a final concentration of 5 mM was added to the extracted protein, which was reduced at 56 °C for 30 min, then IAM with a final concentration of 11 mM was added, and incubated at room temperature in the dark for 15 min. Then, TEAB was added to dilute urea to make its concentration less than 2 M. The tryptic peptides were dissolved by liquid chromatography mobile phase A and separated by EASY-nLC 1200 ultra-high performance liquid system. The mobile phase A was an aqueous solution containing

0.1% formic acid and 2% acetonitrile, while the mobile phase B contains 0.1% formic acid and 90% acetonitrile. Gradient setting: 0-70 min, 6% -22%B; 70.0–83.0 min, 22%-34%B; 83.0–87.0 min, 34%-80%B; 87.0–90.0 min, 80%B, the flow rate is maintained at 500 nL/min. After being separated by the ultra-high performance liquid system, the peptide was ionized by NSI ion source, and then analyzed by Orbitrap Exploreris 480 (Thermo Fisher Scientific) mass spectrometry. The ion source voltage was set to 2.2 kV, and the FAIMS compensation voltage (CV) was set to -45V; −65V. The parent ion of peptide and its secondary fragments were detected and analyzed by high-resolution Orbitrap. The scanning range of the primary mass spectrum was set to 400–1200 m/z, and the scan resolution was set to 60,000 and the fixed starting point of the scanning range of secondary mass spectrum was 110 m/z, the secondary scan resolution was set to 15,000, and TurboTMT was set to Off. Data acquisition mode adopts data-dependent scanning (DDA) program, that is, after the primary scanning, the parent ions of the first 25 peptide segments with the highest signal intensity were selected to enter the HCD collision cell in turn for fragmentation with 28% fragment energy, and at the same time, secondary mass spectrum analysis was carried out in turn. In order to improve the effective utilization rate of mass spectrometry, the automatic gain control (AGC) was set to 100%, the signal threshold was set to 5E4 ions/s, the maximum injection time was set to auto, and the dynamic exclusion time of tandem mass spectrometry scanning was set to 25 s, so as to avoid repeated scanning of parent ions. The resulting MS/MS data was processed using Proteome Discoverer search engine (v2.4.1.15). The database was Mus_musculus_10090_SP_20220107.fasta. Protein, peptide, and PSM identification of FDR was set to 1%. Proteins were quantified by intensity-based absolute quantification (iBAQ). According to the quantitative value of iBAQ intensity, the protein lists of two or more samples with intensity value in the FAST group and zero samples with intensity value in Fed group were screened, and the protein lists with FC > 1.5 and $P$-value < 0.05 were screened. The proteins from the two lists were upregulated differential proteins, and vice versa.

## Transmission electron microscopy imaging

For APEX2 labeled electron microscopy imaging, HepG2 was transfected with indicated plasmid (pLVX-APEX2, pLVX-LBP-APEX2), and was prepared for imaging as previously described[49]. In short, cells were fixed with 2% glutaraldehyde in the buffer (0.1 M sodium cacodylate containing 2 mM $CaCl_2$, pH 7.4) and placed on ice for 45 min. The cells were washed with chilled buffer for $3 \times 3$ min, and then the chilled buffer containing 20 mM glycine was added to quench the unreacted glutaraldehyde. Another $3 \times 3$ min wash with chilled buffer was followed. Next, using a fresh solution containing 1 mg/ml DAB (Sigma) and 5.88 mM $H_2O_2$ in chilled buffer resuspend the cells for 30 min. Thereafter, the cell washing was repeated as described above. For post-fixation staining, chilled 1%(w/v) osmium tetroxide in 0.1 M cacodylate buffer was added to the cells on a shaker for 40 min. Cells were washed as above, and then incubated overnight in a chilled 1% uranyl acetate aqueous solution. The samples were dehydrated in a cold-graded ethanol series (50%, 70%, 90%, and 100%) for 2 min, washed once with anhydrous ethanol at room temperature to avoid possible condensation, soaked in Epon resin using 1:1 (v/v) anhydrous ethanol and resin for 30 min, 100% resin $2 \times 1$ h, and placed into fresh resin to polymerize at 60 °C for 48 h. Use an ultramicrotome (Leica EM UC 7) to cut ultrathin slices with a thickness of 70 nm. The samples were absorbed on a TEM copper grid, and then the electron micrograph was recorded by 120 kV Tecnai T12 transmission electron microscope (FEI). For liver tissue, the DAB and $H_2O_2$ steps were removed and further examined under transmission electron microscopy using standard operating procedures[50,51]. Electron microscopy images were taken using the TEM Imaging & Analysis software (V.4.7 SP1).

## Immunofluorescence staining

For fluorescence imaging experiments, all primary and secondary antibodies were dissolved in 1% BSA in TBST, and the dilution rate was according to the manufacturer's instructions. For fluorescence imaging of Bodipy, the medium was changed after the plasmid was transfected for 24 h, and then Bodipy 558/568 C12 with a final concentration of 1 μM or β-Bodipy FL C12-HPC with a final concentration of 2 μM was added for staining after 16-h of treatment. Both bodipy 665/676 and bodipy 493/503 had a final concentration of 500 ng/ml and were added simultaneously with the secondary antibody to avoid the same fluorescence or cross-staining. Immunofluorescence staining of liver tissue was performed as described previously[52]. Cellular or tissue sections are fixed with 4% paraformaldehyde for 10 min and then rinsed with 1× PBS for three times, each time lasting 5 min. Subsequently, the samples are permeabilized using 0.1% Triton X-100 in PBS for an additional 10 min. Following three more PBS washes, primary antibodies are applied to incubate overnight at 4 °C. After washing with 1× PBS three times, secondary antibodies are added and incubated at room temperature in the dark for 1 h. An additional three PBS washes are performed, followed by the addition of 5 μg/ml DAPI for a 5-min incubation at room temperature. The sections are then washed again with 1× PBS three times and sealed with a fluorescence-preserving mountant. The samples were photographed with ZEISS 800 microscope in airyscan mode, and data were collected by ZEN 2.6 (blue edition).

## ROS detection

The level of ROS in cells was determined by measuring the oxidative conversion of cell permeability DCFH-DA to fluorescent dichloro-fluorescein (DCF) using the Reactive Oxygen Species Assay Kit (Beyotime). According to the manufacturer's instruction, the HepG2 cells were transfected with LBP-Cherry plasmid for 24-h, and the medium was changed, followed by the addition of 1 μM Bodipy C12 for 16-h. Cells were washed with PBS for three times and finally placed into PBS for 2-h. ROS Fluorescence intensity was measured using SpectraMax iD3 (Molecular Devices).

## Hepatic VLDL release assay

The VLDL export assay was measured as previously reported[53]. In brief, after 24 h of starvation, blood was taken from the orbital vein and 1 g/kg poloxamer 407 was injected intraperitoneally. 4 h later measured plasma triglyceride and VLDL concentrations.

## Lipidomic analysis

We performed lipidomic analysis on $LBP^{-/-}$, $LBP^{KI/KI}$, and their WT control mice fed a high-fat diet for 16 weeks, with three biological replicates per group, and the analysis was conducted by APTBIO Inc. For the LBP binding lipidomics analysis, we performed as reported previously[54]. We overexpressed LBP in HepG2 cells for 48 h and then treated the cells with 500 μM $H_2O_2$ for 3 h. After the LBP immunoprecipitation, lipidomic analysis was carried out by APTBIO Inc. The non-targeted lipid analysis platform based on UPLC-Orbitrap mass spectrometry system combined with LipidSearch software (V.4.2, Thermo Scientific) was used for lipid identification and data preprocessing, so as to obtain the relative content of lipid molecules in the samples in large-scale. In brief, the tissues were ultrasonicated in MTBE methanol solution, centrifuged, and then the organic phase was dried with nitrogen. After re-dissolution with 90% isopropanol/acetonitrile solution, the supernatant was collected by centrifugation for detection. The samples were separated using a UHPLC Nexera LC-30A ultra-performance liquid chromatographic system. C 18 column at 45 °C and flow rate of 300 μL/min. Mobile Phase Composition A: aqueous acetonitrile (acetonitrile: water = 6:4, v/v) and B: acetonitrile in 2-propanol (acetonitrile: 2-propanol = 1:9, v/v). The gradient elution procedure

was as follows: 0–2 min, B was maintained at 30%; 2–25 min, B changed linearly from 30% to 100%; 25–35 min, B maintained at 30%. After UHPLC separation, mass spectrometric analysis was performed using a Q Exactive mass spectrometer (Thermo Scientific). ESI source conditions were as follows: Heater Temp 300 °C, Sheath Gas Flow rate 45 arb, Aux Gas Flow Rate15 arb, Sweep Gas Flow Rate 1arb, Spray voltage 3.0KV, Capillary Temp 350 °C, S-Lens RF Level 50%, MS1scan ranges: 200–1800. The mass-charge ratio of lipid molecules and lipid fragments were obtained by collecting 10 fragment maps (MS 2scan, HCD) after each full scan. The resolution of MS1 at M/Z 200 was 70,000 and that of MS2 at M/Z 200 was 17,500. LipidSearch was used for peak identification, peak extraction, and lipid molecules identification (secondary identification). The main parameters were precursor tolerance: 5 ppm, Product Tolerance: 5 ppm, and Product Ion Threshold: 5%. The obtained data were subjected to quality control for subsequent data lipid difference analysis.

### Lipid-binding assay

Lipid molecules Hexenoic acid (S25963), Undecylenic acid (B25313), Heptadecenoic acid (B25734), Eicosapentaenoic acid (B26385), Cervonic acid (B27406), Triheptadecenoin (B24616), Tridocosahexaenoin (B23660) were purchased from the Yuanye Biotechnology Company (Shanghai, China). The lipid-binding assay was performed by following the procedure described previously[55]. In brief, spotted 1 μg of the lipid dilution onto the nitrocellulose membrane, followed by 1 h air dry. The lipid-bound membrane was treated with 1 mM $H_2O_2$ for another 1 h. After a 3 times 5 min wash of TBST, the membranes were blocked in 3% BSA in TBST for 1 h and incubated with 1 μg/mL LBP-His fusion protein at 4 °C overnight. Wash the membrane 10 times over 50 min in TBST. As in the western blot assay, the membrane was incubated with the primary antibody for 1-h at room temperature and then with the HRP-coupled secondary antibody for 1-h. During which the membrane was washed at least 10 times each time, ECL reagent (Biosharp, China) was added, and visualizing bands used a chemiluminescence imager (SH-523, SHST, Hangzhou, China).

### Microscale thermophoresis assay

Purified LBP-His was purchased from MCE (CAS: HY-P70313) and was labeled by NanoTemper Monolith NT™ Protein Labeling Kit RED-NHS 2nd Generation (#MO-L011) according to the manufacturer's procedure. The MST assay was previously described[56]. The lipid molecules Tridocosahexaenoin and LBP-His samples were gradient mixed and loaded into premium capillaries (NanoTemper Technologies). Binding affinities were measured using the Monolith NT.115 device with MO.Control (V.1.6.1) software. Data analyses were performed using MO.Affinity Analysis V.2.3 software (NanoTemper Technologies).

### Lipolysis assay

The lipolysis assay was performed as reported previously[57]. Twenty-four hours after transfection, cells were washed with PBS and the culture media was changed to complete growth media containing 1 μM Bodipy C12, and cultured for 1 day. Cells were serum starved for 2 h and placed in KRBH buffer (30 mM HEPES, 120 mM NaCl, 4 mM $KH_2PO_4$, 1 mM $MgSO_4$, 0.75 mM $CaCl_2$, and 10 mM $NaHCO_3$) with 2% fatty acid-free BSA and 5 mM glucose. 100 nM isoprenaline was used to induce lipolysis. Culture media was collected, and cells were lysed by 1%Triton, TG, and FFAs were measured by commercial kits and normalized to protein concentration.

### Seahorse assay

The cellular oxygen consumption rate (OCR) and fatty acid oxidation (FAO) rate were measured using the Seahorse XFe 96 analyzer (Agilent) according to the manufacturer's instructions. Briefly, cells were divided into 6 cm plates and transfected with plasmids for 24 h. $1 \times 10^4$ cells/well were seeded in the 96-well XF cell culture microplate for

24 h, and the Cell Mito Stress Test Kit was used for the OCR measurement. For FAO rate measurement, $1 \times 10^4$ cells/well were seeded in the 96-well XF cell culture microplate and cultured in growth medium with or without 1 μM Bodipy C12 for 24 h. Then, the culture medium was replaced with substrate-limited DMEM supplemented with 0.5 mM glucose, 1 mM glutamax, 0.5 mM carnitine, 1% FBS. After a further 24 h, the medium was changed to FAO assay medium (KHB buffer:111 mM NaCl, 4.7 mM KCl, 1.25 mM $CaCl_2$, 2 mM $MgSO_4$, 1.2 mM $NaH_2PO_4$ supplemented with 2.5 mM glucose, 0.5 mM carnitine and 5 mM HEPES, adjusted to pH 7.4 at 37 °C) and the final concentration of BSA-conjugated palmitate was 50 μM. Data analysis was performed by the Seahorse XF Cell Mito Stress Test Report Generator package of XFE software (V2.6.1.56).

### Co-IP and Western blot

The total protein was extracted by 1% Triton and sonication. Protein A/G Magnetic Beads (Bimake) were used for Co-IP assay and performed as described previously[58]. After boiling, the supernatant was subjected to SDS-PAGE and transferred to nitrocellulose membranes. Then, blocking with 5% skim milk, membranes were incubated with primary antibodies followed by HRP-conjugated secondary antibodies. ECL reagent (US Everbright) was added, and bands were visualized using a chemiluminescence imager (SH-523, SHST, Hangzhou, China) with SHST Capture software (V.2.0.1.107). The dilution of antibodies is performed according to the manufacturer's instructions, and this is introduced in the "Materials" section.

### Ab initio modeling, molecular docking, and molecular dynamics simulations

The predicted structures of LBP were generated by I-TASSER (V5.2)[59]. The protonation state of all the compounds was set at pH = 7.4, and the compounds were expanded to 3D structures using Open Babel (V2.3.1)[60]. AutoDock Tools (ADT3) were applied to prepare and parametrize the receptor protein and ligands. The docking grid documents were generated by AutoGrid of sitemap, and AutoDock Vina (V1.2.0) was used for docking simulation[61]. The optimal pose was selected to analysis interaction. Finally, the protein-ligand interaction figure was generated by PyMOL(V 2.5). The LBP protein is represented as a slate cartoon model, ligand is shown as a cyan stick, and their binding sites are shown as magentas stick structures. Nonpolar hydrogen atoms are omitted. The hydrogen bond, ionic interactions, and hydrophobic interactions are depicted as yellow, magentas, and green dashed lines, respectively.

The molecular dynamics simulations were carried out with Desmond/Maestro noncommercial version 2022.1 as a molecular dynamic's software[62]. TIP3P water molecules were added to the systems, which were then neutralized by 0.15 M NaCl solution. After minimization and relaxation of the system, the production simulation was performed for 100 ns in an isothermal-isobaric ensemble at 300 K and 1 bar. Trajectory coordinates were recorded every 100 ps. The molecular dynamics analysis was performed using Simulation Interaction Diagram from Desmond.

### Statistical analysis

All biochemical experiments were repeated at least three times to obtain the unequivocal results. The significance of differences between experimental groups were determined using GraphPad Prism 9.4.1. Quantitative data of normal distribution were described with mean ± standard deviation. Data were analyzed by two-tailed Student's $t$-test for comparisons of two samples, one-way ANOVA with Dunnett's and Tukey's post-test for univariate comparisons, two-way ANOVA with Bonferroni's post-test for bivariate comparisons, and the Pearson coefficient for the linear correlation between two variables. Parameters can be found in the legend, where values less than 0.05 were considered statistically significant.

## Reporting summary

Further information on research design is available in the Nature Portfolio Reporting Summary linked to this article.

## Data availability

The RNA-seq data have been made available at the NCBI SRA repository under accession number PRJNA939362. The mass spectrometry proteomics data have been deposited to PRIDE database under accession number PXD040940. Metabolic data were uploaded to Metabolomics Workbench under accession number ST002522. Source data are provided with this paper.

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

## Acknowledgements

This research was supported by the National Science Foundation of China grants 81971875 (H.S.F.), 82071590 (A.D.L.), the Fundamental Research Funds for the Central Universities WK9110000194 (M.Z.), Special Project for Local Science and Technology Development Guided by Anhui Province 2017070802D147 (S.D.Y.) and Talent Training Program, School of Basic Medical Sciences, Anhui Medical University 2022YPJH102 (H.S.F.).

## Author contributions

Conceptualization: H.S.F., Q.L.Z. Methodology: Q.L.Z., H.S.F., X.T.S., M.Z., A.Y.H., C.W.S., J.W., C.C., F.L., H.L. Investigation: Q.L.Z., X.T.S., X.Y., J.H., F.H., J.N., Y.L.Z., Y.G.F., H.B.W., Q.W., T.T.Z., T.Z., L.L.M. Visualization: Q.L.Z., X.T.S., H.S.F. Funding acquisition: H.S.F., S.D.Y., M.Z., A.D.L., Q.S.H. Project administration: Q.L.Z., H.S.F. Supervision: H.S.F., M.Z., S.D.Y., A.D.L. Writing-original draft: Q.L.Z., H.S.F. Writing-review & editing: Q.L.Z., H.S.F., X.T.S., X.Y., M.Z., S.D.Y., M.C.

## Competing interests

The authors declare no competing interests.

## Additional information

[1]Laboratory of Diabetes, Department of Endocrinology, The First Affiliated Hospital of USTC, Division of Life Sciences and Medicine, University of Science and Technology of China, Hefei, Anhui 230001, China. [2]Department of Pathophysiology, Anhui Medical University, Hefei, Anhui 230000, China. [3]Department of Pathology, the First Affiliated Hospital of USTC, Division of Life Sciences and Medicine, University of Science and Technology of China, Hefei, Anhui 230001,

China. [4]Department of pathology, The First Affiliated Hospital of Anhui University of Chinese Medicine, Hefei, Anhui 230011, China. [5]Department of Epidemiology and Biostatistics, School of Public Health, Anhui Medical University, Hefei, Anhui 230032, China. [6]School of Life Sciences, Anhui Medical University, Hefei, Anhui 230000, China. [7]Department of Chemistry, Center for BioAnalytical Chemistry, Hefei National Laboratory of Physical Science at Microscale, School of Life Sciences, University of Science and Technology of China, Hefei, Anhui 230022, China. [8]Department of Hepatobiliary Surgery, The First Affiliated Hospital of USTC, Center for Advanced Interdisciplinary Science and Biomedicine of IHM, Division of Life Sciences and Medicine, University of Science and Technology of China, 230001 Hefei, China. [9]State Key Laboratory of Proteomics, Beijing Proteome Research Center, National Center for Protein Sciences (Beijing), Beijing Institute of Lifeomics, Beijing 102206, China. [10]Organ Transplantation Center, Department of General Surgery, First Affiliated Hospital of Anhui Medical University, Hefei, Anhui 230022, China. [11]Department of Respiratory and Critical Care Medicine, First Affiliated Hospital of Anhui Medical University, Hefei, Anhui 230022, China. [12]Graduate School of Bengbu Medical College, Bengbu, Anhui 233030, China. [13]Experimental Medicine Center, Tongji Hospital, Tongji Medical College, Huazhong University of Science and Technology, Wuhan, Hubei 430030, China. [14]These authors contributed equally: Qilun Zhang, Xuting Shen, Xin Yuan. ✉e-mail: andingliu@tjh.tjmu.edu.cn; ysd196406@163.com; zhengmao1999@foxmail.com; fanghaoshu@ahmu.edu.cn

