## [Peer Review File · Nature Communications]

Lipopolysaccharide Binding Protein resists hepatic oxidative stress by regulating lipid droplet homeostasisREVIEWER COMMENTS

Reviewer #1 (Remarks to the Author):

This is a sound and interesting study that demonstrated how LBP can promote liver severe steatosis, increasing lipid droplet accumulation on obesogenic conditions. Most importantly, this study also revealed that oxidative stress is required for LBP steatotic effects and point to a protective role of LBP in the prevention of oxidative stress-induced lipid peroxidation.

Major comments

- In line with this study, two previous studies demonstrated that LBP KO (PMID: 27404046) or liver specific LBP gene knockdown (PMID: 36090751) reduced liver lipid accumulation, attenuating lipogenesis in feeding conditions (specifically high fat and high sucrose) in which liver lipogenesis was induced (PMID: 36090751). These studies should be mentioned in introduction and discussed in discussion.
- To further investigate the possible role of LBP on liver lipogenesis in mice after 16 weeks of HFD, additional components of lipogenic pathway pSer79ACC/ACC and pThr172AMPK α /AMPK α protein levels should be analyzed in livers of LBP $^{-/-}$, LBPKI/KI and WT mice.
- Also in agreement with current study, two previous studies suggested a protective role of LBP against oxidative stress in liver (PMID: 36410673; PMID: 33988537). These studies should be mentioned and briefly discussed in discussion.
- In my view, since the relationship between LBP and oxidative stress was sustained from markers subtracted from liver transcriptomic/proteomic and lipidomic data, to strength these findings, stable liver ROS markers, such as protein carbonylation should be measured in livers of LBP $^{-/-}$, LBPKI/KI and WT mice.
- The figures have very low resolution and need to be substantially improved in order to be able to correctly assess the results described, especially in the graphs.
- Which is the impact of 16W HFD on insulin resistance in LBP $^{-/-}$, LBPKI/KI and WT mice? At least circulating markers such as fasting glucose and insulin levels should be measured and shown in results in all experiments, including in mice under chronic jet lag and forced swimming test.
- Seems that the obesogenic effects of 16W HFD was not observed in ketogenic diet. This discrepancy should be explained and discussed in discussion.
- Figure 5d-f showed the effect of NAC in LBP KI/KI mice fed with chow diet, but what is the effect of NAC on body weight of 16W HFD?? Specifically, in the following experiment stated in methods: "In the rescue experiment, the mice were fed 588 with 60% HFD for 7 weeks, and treated with NAC (1g/L in drinking water) from the 4th week". This information is crucial to confirm if antioxidants can prevent LBP-associated obesity.

Reviewer #2 (Remarks to the Author):

In this manuscript, Zhang and coworkers report an interesting relationship between LD homeostasis and oxidative stress via the activity of the Lipopolysaccharide-binding protein.

I will first comment on the methodology, claims and conclusions related to the modeling aspects of the proposed mechanism of TG-binding by LBP, as they fall within my main expertise, and I will then conclude with more general comments on the manuscript.

For what concerns the computational methods described in the paper, they are fairly standard and quite well-established, but the authors 1. do not describe them appropriately and 2. seem to overinterpret them.

For what concerns point 1., it is not clear if they use the AF model or the X-ray structure in their docking protocol as this is not precisely described in the methods section. This is particularly relevant as the authors suggest that part of the structure might be quite mobile, and hence their structural prediction might come with low accuracy. In addition, the authors do not specify if protein and/or ligands are free to rotate during the docking procedure. This might be relevant to distinguish between acyl chains of different saturation level (e.f SFA vs MUFA vs PUFA).

For what concerns point 2, the free energies that the authors obtain using the docking protocol are very similar, and they do not support any further claim. Rather, they might serve as an hypothesis for further lipid binding assays, but these are not shown nor performed. Also, in the methods section the authors describe a lipid binding assay. First the description is clearly insufficient to reproduce the protocol and below the acceptable standard for lipid binding assays. Second, it is unclear to me wherein whether they use this assay in the results section and what is the lipid they want to measure binding for.

Overall, I have similar criticisms for what concerns the entire manuscript. While ambitious, the manuscript falls short repeatedly in two aspects: 1. clarity of the experiments performed (which are often poorly described) and 2. frequent over-interpretation of the presented results. Because of this, I think it falls short of the high standard of Nature Communications.

Reviewer #3 (Remarks to the Author):

I have carefully reviewed the manuscript titled "LBP resists hepatic oxidative stress by regulating LD homeostasis". The authors have delved into an intriguing potential role for the lipopolysaccharide-binding protein (LBP) as an antioxidant, demonstrating its interplay with lipid droplet (LD) metabolism

and redox signaling in hepatocyte lipid peroxidation resistance. Notably, the study underscores the triglyceride capture capacity of LBP, emphasizing its dependency on the C-terminal groove. This insight, in turn, offers a plausible explanation for the LD-translocation of LBP and subsequent enhancement of LD stability. Furthermore, the revelation that the LBP-F436L mutation associates with reduced hepatic TG is of notable significance.

Despite the meticulous design and execution of the study, I recommend the authors to consider the following suggestions to enhance the comprehensiveness and impact of their research:

Glucose and Fatty Acid Metabolism: LD accumulation in the liver may modulate glucose metabolism. Given this premise, it would be insightful for the authors to elucidate the influence of both LBP and its LBP-F436L variant on glucose and fatty acid metabolism. Specifically, a deeper dive into potential alterations in genes associated with gluconeogenesis and lipogenesis would add considerable value.

PLIN2 Association: The relationship between PLIN2 and LBP warrants a more detailed exploration. Providing an in-depth explanation of this association could clarify certain mechanistic aspects of LBP's role in LD homeostasis.

Inflammatory Biomarkers: Given the well-established link between hepatic LD accumulation and progression toward inflammation, it is worth exploring the role of LBP and the LBP-F436L variant on inflammatory pathways. I recommend the authors consider evaluating liver levels of inflammatory cytokines to draw a more comprehensive picture.

LBP-F436L in Humans: Extending the research's applicability to a clinical setting, it would be valuable for the authors to delve into human datasets. Exploring the potential effects of the LBP-F436L variant on the risk of hepatic lipid droplet accumulation and inflammation could pave the way for translational research opportunities.

In conclusion, the research presented in the manuscript offers promising insights into the multifaceted role of LBP in hepatic oxidative stress resistance. By considering the above suggestions, the authors can further solidify their findings and broaden the scope of their study.

Reviewer #4 (Remarks to the Author):

LBP has long been known to elicit immune responses when exposed to LPS. In this manuscript, Zhang et al revealed a novel function of LBP in coupling lipid droplet metabolism and redox signal. The authors showed (oxidative)-triglyceride capture activity of LBP under oxidative stress which promotes the LD growth. Elimination of ROS caused export of LBP from LD via association with PDRX4 to promote lipid synthesis. Overall, this is a very interesting work describing the role of LBP in oxidative stress. Nonetheless, some explanations are not convincing and need improvement.

My other comments:

1. Line 88-90. "We next sought to reveal the molecular basis underlying LBP phase transition with LD, and found LBP contains an intrinsically disordered region (IDR) in N-segment with serine bias (Fig. 2c)."

The authors claimed LBP contains IDR region in the N-segment. This is problematic as the serine-rich region (IDR region) is well-ordered as beta-sheet in both experimental X-ray (PDB 4M4D) and Alpha Fold predicted structures. This will compromise their conclusion too since it's not surprising with so many mutation points.

Besides, the software PONDR VSL2 is not cited.

2. Line 99-101. "We noticed a highly conserved sequence, AA 286-297, was invisible in the 3D structure of LBP, which was predicted to be an α -helix (#4-101 helix, Fig. 2f) using the AlphaFold AI system...".

I couldn't see the evidence for this claim that this sequence is highly conserved.

3. Line 103-105. "Therefore, we mutated all unstable amino acids in the #4-helix into those that were to form a stable α -helix (DYVFNTASLVYHEE to YAAMAMLLALMLAL, 305-316aa were 105 deleted), and observed that this mutation inhibited its translocation to LD (Fig. 2g-h)."

This experiment design is problematic.

First of all, the term of unstable amino acids is not accurate.

In addition, 305-316 aa region covers a conserved beta-strand and part of the antiparallel beta-sheet. I doubt the truncation mutant with deleted 305-316 aa region might become unstable or unfolded, comprising the conclusion of this experiment.

4. Line 125-127. "Thus, we hypothesize that the LBP promotes steatosis through oxidative-TG capture and LD accumulation activities."

It lacks lipidomic data to support its preference towards oxidized TG.

5. Line 205-208. "In fact, phosphatidylcholine (PC) treatment caused LBP to be exported from LD and transported back to ER, revealing that PC contributes to the shuttle-out process of LBP (Extended Data Fig. 8a)."

The modelled PC in the previously reported LBP structure lacks the electron density for the headgroup which is exposed to the solvent, suggesting LBP might bind other lipids too. Have the authors test PE or other lipids in the treatment experiments?

6. Some typos. Line 80 "... mobilize..." to "... mobilized...", Line 189 "... interscts..." to "... interacts...", Line 207 "... contributs..." to "... contributes...", Line 233 "... is responds..." to "... responds...", Line 234 "... remains ..." to "... remains...".

Reviewer #1:

Q1: In line with this study, two previous studies demonstrated that LBP KO (PMID: 27404046) or liver specific LBP gene knockdown (PMID: 36090751) reduced liver lipid accumulation, attenuating lipogenesis in feeding conditions (specifically high fat and high sucrose) in which liver lipogenesis was induced (PMID: 36090751). These studies should be mentioned in introduction and discussed in discussion.

Response:

We greatly appreciate your valuable suggestion. The addition of LBP knockout studies that suppress lipogenesis and lipid accumulation further supports our findings and enhances the persuasiveness of our paper. We have revised the introduction and discussion sections of our paper, mentioned these studies and highlighted them in red.

Q2: To further investigate the possible role of LBP on liver lipogenesis in mice after 16 weeks of HFD, additional components of lipogenic pathway pSer79ACC/ACC and pThr172AMPK α /AMPK α protein levels should be analyzed in livers of LBP^{-/-}, LBP^{KI/KI} and WT mice.

Response:

We appreciate your valuable suggestion. ACC, as a key enzyme in the process of lipid de novo synthesis, is inhibited by phosphorylation at Ser79. Activated p-AMPK α can phosphorylate ACC, inhibiting its activity. We examined the levels of p-ACC/ACC and pThr172AMPK α /AMPK α in the livers of WT, LBP^{KI/KI}, and LBP^{-/-} mice after 16 weeks of HFD and presented our findings in Extended Data Fig.2 E-2G. Consistent with previous studies, knocking out LBP increased the ratio of pSer79ACC/ACC, indicating inhibition of lipid de novo synthesis. LBP^{KI/KI} decreased the ratio of pSer79ACC/ACC, suggesting a promotion of lipid de novo synthesis.

Q3: Also in agreement with current study, two previous studies suggested a protective role of LBP against oxidative stress in liver (PMID: 36410673; PMID: 33988537). These studies should be mentioned and briefly discussed in discussion.

Response:

We greatly appreciate your valuable suggestion. The relationship between LBP and oxidative stress has been reported in several studies. The study you mentioned, PMID: 33988537, is actually a previous work from our lab. At that time, we found that

knocking out LBP caused severe oxidative stress, which motivated us to establish the LBP^{KI/KI} mouse model and conduct the research presented in this article. We have revised the discussion section of our paper, mentioned these studies and highlighted them in purple. We believe these modifications have enhanced the persuasiveness of our article.

Q4: In my view, since the relationship between LBP and oxidative stress was sustained from markers subtracted from liver transcriptomic/proteomic and lipidomic data, to strength these findings, stable liver ROS markers, such as protein carbonylation should be measured in livers of LBP^{-/-}, LBP^{KI/KI} and WT mice.

Response:

We greatly appreciate your valuable input. The original article did not provide sufficient evidence to support the relationship between LBP and oxidative stress, as it relied solely on ROS measurements and lipid probe Bodipy 665/676 data. Therefore, we further measured the levels of protein carbonylation, nitration, and the lipid peroxidation marker MDA in the livers of WT, LBP^{KI/KI}, and LBP^{-/-} mice after 16 weeks of HFD. The results showed a significant antioxidant effect in LBP^{KI/KI} mice. We have added this data to Fig. 4D-4F.

Q5: The figures have very low resolution and need to be substantially improved in order to be able to correctly assess the results described, especially in the graphs.

Response:

We apologize for the low-resolution figures in the initial submission. Due to file size limitations, we had to compress the images excessively. In this round of revisions, we have rearranged the figures and included high-resolution images in the article.

Q6: Which is the impact of 16W HFD on insulin resistance in LBP^{-/-}, LBP^{KI/KI} and WT mice? At least circulating markers such as fasting glucose and insulin levels should be measured and shown in results in all experiments, including in mice under chronic jet lag and forced swimming test.

Response:

We appreciate your valuable input. Incorporating the role of LBP in glucose and lipid metabolism regulation would greatly enhance the logic and persuasiveness of our article. We have added data on fasting glucose and insulin levels in WT, LBP^{KI/KI}, and

LBP^{-/-} mice after 16 weeks of high-fat diet, as shown in Figure 7C-7D. Additionally, due to time constraints, we tested GTT and ITT levels in our existing WT, LBP^{KI/KI}, and LBP^{-/-} mice that had been fed a high-fat diet for 22 weeks. We found that LBP^{KI/KI} mice exhibited significant insulin resistance and glucose tolerance, consistent with previous studies (PMID: 22184060). We also conducted a metabolic cage experiment and added the results to Figure 7P, confirming the shift from glucose to lipid utilization in LBP^{KI/KI} mice. We performed new FST and CJL experiments and found that chronic stress increased fasting glucose and insulin levels and induced insulin resistance in WT mice. The relevant data are presented in Extended Data Fig. 7H-7L.

Q7: Seems that the obesogenic effects of 16W HFD was not observed in ketogenic diet. This discrepancy should be explained and discussed in discussion.

Response:

We appreciate your suggestion. We have added a discussion on why a ketogenic diet does not induce LBP-mediated obesity, highlighting it in blue text. This enhances the integrity and persuasiveness of our article.

Q8: Figure 5d-f showed the effect of NAC in LBP KI/KI mice fed with chow diet, but what is the effect of NAC on body weight of 16W HFD?? Specifically, in the following experiment stated in methods: “In the rescue experiment, the mice were fed 588 with 60% HFD for 7 weeks, and treated with NAC (1g/L in drinking water) from the 4th week” . This information is crucial to confirm if antioxidants can prevent LBP-associated obesity.

Response:

We appreciate your valuable input, and we apologize for the oversight in the initial draft. In fact, we conducted two sets of experiments to study the inhibitory effect of NAC on LBP-mediated obesity. We investigated the primary prevention of NAC administration during HFD, as shown in Figure 7Q of the revised manuscript, which lasted for 16 weeks of HFD. Additionally, we examined secondary prevention, where NAC treatment was given after 7 weeks of HFD and the appearance of obesity symptoms. We found that both primary and secondary prevention with NAC showed a reduction in body weight. These compelling data confirm that NAC can prevent LBP-associated obesity. Due to space limitations, we did not present the secondary prevention data and we have corrected the relevant section in the methods.

Reviewer #2:

Q1: It is not clear if they use the AF model or the X-ray structure in their docking protocol as this is not precisely described in the methods section. This is particularly relevant as the authors suggest that part of the structure might be quite mobile, and hence their structural prediction might come with low accuracy. In addition, the authors do not specify if protein and/or ligands are free to rotate during the docking procedure. This might be relevant to distinguish between acyl chains of different saturation level (e.f SFA vs MUFA vs PUFA).

Response:

We greatly appreciate your constructive criticism. Your expertise has helped us recognize the most significant shortcoming in our work and has provided guidance for our future research direction. In the initial draft, we used the LBP structure obtained from X-ray crystallography (PDB 4M4D) for rigid docking with small molecule lipids. Although the data indicated that the hydrophobic pocket of LBP could bind lipids, this conclusion may be inaccurate.

In this revised manuscript, we have reexamined the structure of LBP. We found that the #4-helix of LBP is highly conserved in mammals and exhibits an α -helix form in both homology modeling and ab initio modeling (Figure 6A-6B). In contrast, the X-ray crystallography-derived LBP structure does not show this helix, possibly due to the absence of lipid ligands during structure determination, leading to a highly flexible region. Consequently, we used the ab initio modeled LBP structure for molecular docking with lipids and allowed both the protein and ligands to rotate freely. By introducing hydroxyl groups to the double bonds at the carboxyl end of the lipid molecules to simulate the oxidized state, we obtained binding energies with differences (Figure 6C-6D). Furthermore, we performed molecular dynamics simulations of the ab initio modeled LBP structure with lipid molecules, revealing that the oxidized small molecules undergo more severe changes upon encountering LBP. Hydrophobic forces, hydrogen bonds, and water bridges are crucial for LBP binding to lipids. Among these, the hydrogen bond formed between His_294 at the C-segment #4-helix of LBP and the oxidized lipid is essential for LBP entry into lipid droplets (Figure 6E-N).

Nevertheless, the exact binding mode of LBP and lipids is still worth further exploration. While we cannot currently obtain high-resolution LBP crystal structures, we have conducted computational simulations and experimental validation to preliminarily confirm the way LBP captures oxidized lipids. In fact, we are already

working on purifying the protein. We will report our subsequent experimental progress in our next article. Thank you again for your constructive criticism.

Q2: The free energies that the authors obtain using the docking protocol are very similar, and they do not support any further claim. Rather, they might serve as an hypothesis for further lipid binding assays, but these are not shown nor performed. Also, in the methods section the authors describe a lipid binding assay. First the description is clearly insufficient to reproduce the protocol and below the acceptable standard for lipid binding assays. Second, it is unclear to me wherein whether they use this assay in the results section and what is the lipid they want to measure binding for.

Response:

We appreciate your criticism and apologize for the oversights in the initial draft. Given the uncertainty of computational simulations, it is essential to verify the binding ability of LBP to lipids and its preference for oxidized lipids experimentally. We have added experimental validation of LBP binding to lipids in the revised manuscript. We used LBP immunoprecipitation to detect the difference in LBP binding to lipids under oxidized and non-oxidized stimuli in HepG2 cells and observed that LBP bound to more lipids under oxidative stress, with a preference for unsaturated and long-chain lipids (Figure 3H-3J). We further conducted an in vitro lipid binding assay using a series of unsaturated fatty acids and triglycerides, combined with microscale thermophoresis (MST) experimental validation, demonstrating the preference of LBP for oxidized long-chain unsaturated lipids (Figure 3K-3L). These experiments provide strong experimental data support for our previous computational simulations, illustrating the preference of LBP for oxidized unsaturated lipids.

Reviewer #3:

Q1: Glucose and Fatty Acid Metabolism: LD accumulation in the liver may modulate glucose metabolism. Given this premise, it would be insightful for the authors to elucidate the influence of both LBP and its LBP-F436L variant on glucose and fatty acid metabolism. Specifically, a deeper dive into potential alterations in genes associated with gluconeogenesis and lipogenesis would add considerable value.

Response:

We thank your valuable suggestion. We agree that lipid accumulation in the liver may influence glucose metabolism. In our revised manuscript, we have included additional studies on the effects of LBP and LBP-F436L mutation on glucose and lipid metabolism. We found that, after 16 weeks of high-fat diet, LBP^{KI/KI} mice had higher fasting blood glucose and insulin levels compared to WT and LBP^{-/-} mice (Figure 7C-7D). We also examined the GTT and ITT levels of WT, LBP^{KI/KI}, and LBP^{-/-} mice fed a high-fat diet for 22 weeks and found that LBP^{KI/KI} mice exhibited significant insulin resistance and glucose tolerance phenotypes (Figure 7E-7H). To investigate the effects of LBP on glucose and lipid metabolism, we performed a series of Seahorse assays in HepG2 cells with LBP overexpression (Figure 7J-7O), demonstrating that overexpression of LBP led to a preference for lipid-based energy and suppressed glucose consumption. For the F436L, we also conducted corresponding validation but did not present the results in the article. Overall, we found that the F436L variant acted as a loss-of-function mutation, with its effects falling between the control group and the WT LBP group, as shown in the figure below.

For lipid synthesis, ACC plays a key role in de novo lipogenesis. Phosphorylation of ACC at Ser79 inhibits its activity. Activated p-AMPK α can phosphorylate ACC, inhibiting its activity. We measured the levels of p-ACC/ACC and pThr172AMPK α /AMPK α in WT, LBP^{KI/KI}, and LBP^{-/-} mice after 16 weeks of high-fat feeding and presented the results in Extended Data Fig.2E-2G. Consistent with previous studies, knocking out LBP increased the ratio of pSer79ACC/ACC in the liver, indicating inhibition of de novo lipogenesis. LBP^{KI/KI} decreased the pSer79ACC/ACC ratio,

suggesting promotion of de novo lipogenesis. For gluconeogenesis, we did not identify differentially expressed proteins related to gluconeogenesis, such as PEPCK and G6PC, in the proteomic data. This may be due to the compensatory glycogen breakdown in LBP^{KI/KI} mice (Figure 7I).

Q2: PLIN2 Association: The relationship between PLIN2 and LBP warrants a more detailed exploration. Providing an in-depth explanation of this association could clarify certain mechanistic aspects of LBP's role in LD homeostasis.

Response:

We appreciate your valuable suggestion. PLIN2, as one of the most abundant proteins on lipid droplets, has been shown to influence lipid droplet dynamics through interactions with other proteins and enzymes (PMID: 30523332). We have previously investigated the relationship between LBP and PLIN2. However, our APEX proximity labeling pull-down experiments and Co-IP experiments demonstrated that LBP does not bind to PLIN2. In addition, as shown below, we found that the level of PLIN2 in lipid droplets of LBP^{KI/KI} mice was higher than that of WT mice, which may further inhibit lipolysis, consistent with the observations in this study. Due to our current data, we do not understand the specific molecular mechanism by which LBP^{KI/KI} mediates the increased lipid droplet localization of PLIN2. Therefore, we did not present this in the article. The regulatory role of LBP on the PLIN family requires further exploration in future studies.

Q3: Inflammatory Biomarkers: Given the well-established link between hepatic LD accumulation and progression toward inflammation, it is worth exploring the role of LBP and the LBP-F436L variant on inflammatory pathways. I recommend the authors consider evaluating liver levels of inflammatory cytokines to draw a more comprehensive picture.

Response:

We appreciate your constructive suggestion. Inflammation plays a crucial role in promoting liver LD accumulation, which occurs through two main pathways: one is by activating IKK β , leading to the degradation of I κ B and the release of NF- κ B into the nucleus, promoting the transcription of inflammatory-related genes; the other is by phosphorylating JNK, further activating AP1, and the phosphorylated AP1 enters the nucleus, inducing the transcription of inflammatory-related genes. In this revised manuscript, we have validated our findings using WT, LBP^{KI/KI}, and LBP^{-/-} mice fed a high-fat diet for 16 weeks. We found that the ratio of p-I κ B/I κ B and p-JNK/JNK was significantly increased in LBP^{KI/KI} mice (Extended Data Fig.2H-2J). Additionally, we conducted qPCR experiments to detect the expression of inflammatory-related genes in the livers of WT, LBP^{KI/KI}, and LBP^{-/-} mice fed a high-fat diet for 16 weeks, and found that IL6 and other inflammatory factors were significantly upregulated in LBP^{KI/KI} mice (Extended Data Fig.2K). These findings suggest that LBP^{KI/KI}-induced LD accumulation is associated with the activation of inflammatory pathways. Based on our current study, we believe that LBP-induced LD accumulation in the liver is affected by a long time high-fat diet and exhibits a significant phenotype after 16 weeks of high-fat feeding. For the validation of the F436L mutation, creating an endogenous mouse model for this specific mutation would indeed require a significant investment of time and resources, particularly when it comes to the 16-week high-fat feeding regimen. Exogenous overexpressing LBP through viral or high-pressure water jet delivery methods may not be the most suitable approach for long-term studies, as it is difficult to maintain a consistent expression level throughout the high-fat feeding period. Nevertheless, we have demonstrated that LBP^{KI/KI} significantly increased liver inflammatory factor levels after high-fat feeding, providing us with a comprehensive understanding of LBP-induced fatty liver.

Q4: LBP-F436L in Humans: Extending the research's applicability to a clinical setting, it would be valuable for the authors to delve into human datasets. Exploring the potential effects of the LBP-F436L variant on the risk of hepatic lipid droplet accumulation and inflammation could pave the way for translational research opportunities.

Response:

We appreciate your valuable suggestions. Exploring the clinical data of the F436L variant's effects on hepatic lipid droplet accumulation and inflammation would enhance the clinical significance of this article. We analyzed the clinical significance of the F436L mutation through the dbSNP database (<https://www.ncbi.nlm.nih.gov/snp>) and the GeneAtlas database (<http://geneatlas.roslin.ed.ac>). We found that the

frequency of the F436L mutation is very low. Although there is currently a lack of large-scale clinical studies on MAFLD and SNPs, and GeneAtlas does not have a classification related to MAFLD, we still saw a large number of MAFLD-related entries in the diseases related to F436L, as shown in the table below.

Variant	Position	Eff. Allele	Trait	p-value	MAF	ORbeta*	HWE
rs2232618	37001761	T	Body mass index (BMI)	0.0001864	0.08767	-	0.3231
rs2232618	37001761	T	Trunk fat percentage	0.00060967	0.08767	-	0.3231
rs2232618	37001761	T	Trunk fat mass	0.0011287	0.08767	-	0.3231
rs2232618	37001761	T	Whole body fat mass	0.00053767	0.08767	-	0.3231
rs2232618	37001761	T	I10 Essential (primary) hypertension	0.00054328	0.08767	1.03	0.3231
rs2232618	37001761	T	I10-I15 Hypertensive diseases	0.00070884	0.08767	1.03	0.3231
rs2232618	37001761	T	Waist circumference / Hip circumference	0.10856	0.08767	-	0.3231
rs2232618	37001761	T	Weight	0.0015112	0.08767	-	0.3231
rs2232618	37001761	T	K75 Other inflammatory liver diseases	0.16878	0.08767	0.881	0.3231
rs2232618	37001761	T	K70-K77 Diseases of liver	0.10125	0.08767	0.946	0.3231

Regarding the anti-inflammatory effect, recent articles have reported that F436L reduces the binding ability of LBP to LPS, thereby exerting an anti-inflammatory effect (PMID: 34921059). The anti-inflammatory effect of F436L in metabolism has not been reported.

Reviewer #4:

Q1: Line 88-90. “We next sought to reveal the molecular basis underlying LBP phase transition with LD, and found LBP contains an intrinsically disordered region (IDR) in N-segment with serine bias (Fig. 2c).”

The authors claimed LBP contains IDR region in the N-segment. This is problematic as the serine-rich region (IDR region) is well-ordered as beta-sheet in both experimental X-ray (PDB 4M4D) and Alpha Fold predicted structures. This will compromise their conclusion too since it’s not surprising with so many mutation points. Besides, the software PONDR VSL2 is not cited.

Response:

We greatly appreciate your insightful comments, as they have helped us recognize a critical error in our prediction of the IDR region: we failed to compare our predicted data with the LBP structure, thus overlooking the fact that this region covers the beta-sheet structure area. Although we observed significant changes after mutating serine residues, the numerous mutation sites and the alteration of protein rigidity structure cannot accurately validate our hypothesis on LBP phase separation.

Therefore, we have removed the relevant content from the original manuscript. In fact, the mobility of LBP in cells does exist (Extended Data Fig. 6B). Our study shows that the mobility of LBP is more likely due to its interaction with lipids. We conducted molecular dynamics simulations, demonstrating that hydrophobic forces play a crucial

role in the interaction between LBP and lipids (Figure 6K). Consequently, we observed that the interaction between LBP and lipid droplets weakened and tended to diffuse in the cell after disrupting hydrophobic forces with 1,6-hexanediol (Extended Data Fig. 6A). Thank you again for your valuable input, which has allowed us to recognize our incorrect interpretation of the observed phenomena in our study.

Q2: Line 99-101. “We noticed a highly conserved sequence, AA 286-297, was invisible in the 3D structure of LBP, which was predicted to be an α -helix (#4-101 helix, Fig. 2f) using the Alphafold AI system...” .

I couldn't see the evidence for this claim that this sequence is highly conserved.

Response:

We greatly appreciate your valuable input. In the revised manuscript, we have more clearly shown the sequences and secondary structures of the #4-helix in different species, demonstrating the conservation of this alpha helix (Figure 6B).

Q3: Line 103-105. “Therefore, we mutated all unstable amino acids in the #4-helix into those that were to form a stable α -helix (DYVFNTASLVYHEE to YAAMAMLLALMLAL, 305-316aa were 105 deleted), and observed that this mutation inhibited its translocation to LD (Fig. 2g-h).”

This experiment design is problematic.

First of all, the term of unstable amino acids is not accurate. In addition, 305-316 aa region covers a conserved beta-strand and part of the antiparallel beta-sheet. I doubt the truncation mutant with deleted 305-316 aa region might become unstable or unfolded, comprising the conclusion of this experiment.

Response:

We greatly appreciate your valuable input. We must admit that this truncation mutation experiment design was unscientific. The modification of a large portion of the protein, particularly the rigid region, to verify the importance of the #4-helix for lipid binding could potentially impact protein function and lead to unreliable results.

Therefore, we have removed the relevant content from the revised manuscript and changed our strategy for validating the importance of the #4-helix. We confirmed the existence of the #4-helix through secondary structure prediction, homology modeling, and de novo modeling. We then performed molecular dynamics simulations, to identify the His_294 residue in the #4-helix as crucial for forming hydrogen bonds with oxidized lipids and facilitating LBP entry into lipid droplets (Figure 6E-N).

Q4: Line 125-127. “Thus, we hypothesize that the LBP promotes steatosis through oxidative-TG capture and LD accumulation activities.”

It lacks lipidomic data to support its preference towards oxidized TG.

Response:

We appreciate your valuable input. Providing lipidomic data to support the preference of LBP for oxidized TG would significantly strengthen the persuasiveness of our study. We conducted LBP immunoprecipitation experiments on HepG2 cells treated with and without H₂O₂, following by lipid extraction and lipidomics analysis. Due to the instability of oxidized lipids, our data only showed that LBP bound to more triglycerides after H₂O₂ treatment and had a preference for unsaturated and long-chain lipids (Figure 3H-3J). Additionally, we further confirmed the preference of LBP for oxidized TG through in vitro lipid loading and microscale thermophoresis (MST) assay (Figure 3K-3L).

Q5: Line 205-208. “In fact, phosphatidylcholine (PC) treatment caused LBP to be exported from LD and transported back to ER, revealing that PC contributes to the shuttle-out process of LBP (Extended Data Fig. 8a).”

The modelled PC in the previously reported LBP structure lacks the electron density for the headgroup which is exposed to the solvent, suggesting LBP might bind other lipids too. Have the authors test PE or other lipids in the treatment experiments?

Response:

Thank you for your insightful suggestion. We used LBP immunoprecipitation to detect the difference in LBP binding to lipids under oxidized and non-oxidized stimuli in HepG2 cells. The lipidomics data showed that LBP indeed binds to PE and other lipids, and the binding level to PE decreases after oxidative stress. Additionally, we demonstrated strong interactions between LBP and PI, PA using Membrane Lipid Strip from Echelon Biosciences. This may imply the crucial role of LBP in signal transduction, which is worthy of exploration in future studies.

Q6: Some typos. Line 80 “... mobilize...” to “... mobilized...”, Line 189 “... interscts...” to “... interacts...”, Line 207 “... contributes...” to “... contributes...”, Line 233 “... is responds...” to “... responds...”, Line 234 “... remains ...” to “... remains...” .

Response:

Thank you for pointing out the typos. We apologize for the oversight and have corrected the errors in the revised manuscript.

REVIEWER COMMENTS

Reviewer #1 (Remarks to the Author):

All my previous comments have been fully and satisfactorily addressed and in my view the conclusions are strongly supported by the data presented throughout the study.

I did not have further comments.

Reviewer #2 (Remarks to the Author):

I appreciate the author's efforts to address my concerns (and those of the other reviewers). Overall, while the authors identify an interesting relationship between lipopolysaccharide-binding protein (LBP), lipid droplet (LD) metabolism and redox signaling, the new simulations and experiments do not allow to clarify the mechanism behind the role of LBP in oxidative stress conditions. Rather, the mechanistic interpretation of the experiments is insufficient and they confirm my initial assessment that the manuscript has two major issues: 1. clarity of the experiments performed (which are often poorly described and analyzed) and 2. frequent over- and misinterpretation of the presented results.

Specifically:

- The authors report lipid binding experiments using two distinct in vitro assays, and they interpret them as confirming their original hypothesis emerging from the immunoprecipitation assay coupled with lipidomics that LBP binds preferentially to oxidized TGs (the immunoprecipitation experiments rather shows preference for polyunsaturated TGs though...). However, while the first assay (lipid strips), shows enhanced binding of TG to LBP after H₂O₂ treatment, the second one, microscale thermophoresis (MST), shows the opposite trend, with a micromolar K_d for oxidized TG vs nanomolar for non-oxidized TG. However, the authors misinterpret the MST experiment and do not discuss this discrepancy. One alternative explanation of their data (based on the MST experiments) is that the enhanced expression of LBP is a consequence of decreased binding to TG molecules under oxidative stress.
- The docking and MD simulations reported are inconclusive and likely not statistically significant. Specifically, the high RMSF and RMSD of the ligand is likely to indicate unstable binding. A quantification of the protein-ligand interaction over time (e.g. RMSD, distance between the ligand and specific protein residues) would provide more information about the stability of the binding poses found by the docking protocol.
- The subcellular localization of LBP is very odd and would require further characterization: LBP is a secreted protein, and it contains non-hydrophobic domains. Yet, under specific conditions, the authors show its localization inside intracellular LDs. Since LDs are fat deposits, it is unclear how a soluble protein could be found at these cellular sites. While other proteins have been proposed to potentially reside

inside LDs, this is very controversial and it has, to my understanding, never been shown for other soluble proteins of the LBP family (SMP domains).

- Several sentences in the text are not supported by the data shown, for example “To explore how LBP interacts with PRDX4, we simulated the binding mode of LBP and PRDX4 using molecular docking, which revealed that the N-segment of LBP had a high binding affinity for PRDX4 (Extended Data Fig. 5G)”  no quantification of the binding affinity is shown in Ext Data Fig 5G; “Additional experiments demonstrated LBP H294A and H294G mutations substantially impaired LBP-lipid binding capacity (Figure 6N)”  the experiment shown in Figure 6N is not a lipid binding experiment

Minor point: “small molecule 11094-59-0” should be “Tridocosaheptaenoic acid”

Reviewer #4 (Remarks to the Author):

The authors have addressed all my comments. In the revision, they corrected their mistake on IDR, and provided evidence on the #4-helix (sequence alignment, simulation etc). They also provided some evidence for preference of LBP towards oxidized TG.

I have only one minor comment on Fig6.

Figure 6a and 6b, the secondary structure features are in different colour scheme. Helix in 6a is in yellow while in 6b, beta-strands are coloured yellow.

To review #2:

Thank you for providing us with insightful and constructive feedback on our manuscript. Your expertise has significantly enhanced the quality of this manuscript. We have carefully considered your comments and have provided a detailed response. We apologize for any oversights on our description. If there are still any lingering concerns, we would be more than willing to provide further clarification or make necessary revisions.

Q1:

We sincerely apologize for the oversight in the MST result. To clarify, due to the use of different ligand concentrations in our MST assay, we were unable to combine the data for analysis. As a result, only one set of data was presented in our manuscript. Moreover, during the data sorting process, we made inadvertent errors in the labeling of certain samples due to inconsistent color coding provided by the software used. To ensure the accuracy of our findings, we rigorously repeated the MST experiments and conducted a merging replicates analysis. As a result, we obtained consistent conclusions in line with our previous research. The updated results can be referred to in Figure 3L.

Q2:

Thank you for your valuable feedback. In order to enhance the overall data quality of the article, we have removed the MD-related data from the manuscript.

Q3:

The localization of LBP within intracellular lipid droplets (LDs) was indeed an unexpected finding for us as well. Initially, we were skeptical about the presence of polar molecules inside LDs. While SMP domains has been associated with their anchoring between different organelle membranes to form lipid transport channels, the function of LBP appears to be different. To address this intriguing observation, we employed various techniques including electron microscopy, high-resolution fluorescence imaging, and LD isolation analysis. Encouragingly, we consistently observed positive signals exclusively within the interior of LDs, rather than on the LD membranes. These localization studies led us to speculate that LBP might possess lipid binding activity, facilitating its entry into LDs. However, further investigations are required to fully elucidate the structural aspects of LBP in hydrophobic environments, and this is an area we are actively pursuing in our future studies.

In addition, to demonstrate the lipid-binding ability of LBP, we conducted a new in vitro experiments to verify the interaction between LBP and TG, as shown in the figure below.

LBP was fluorescently labeled and purified using the NanoTemper Monolith NT™ Protein Labeling Kit RED-NHS 2nd Generation (#MO-L011), and then mixed extensively with TG. Following a 3-hour incubation period, we observed that LBP, acting as an emulsifier, effectively facilitated the dispersion of TG in the aqueous phase. Remarkably, the presence of LBP was detected not only in the aqueous phase but also in the oil phase. These significant findings strongly substantiate the interaction between LBP and lipids.

Q4:

We apologize for any confusion related to the statements you mentioned.

1. In our study, we used the ClusPro software^[1] to perform molecular docking simulations of the LBP-PRDX4 interaction. The result shown in Extended Data Fig. 5G represents the lowest energy model among the equilibrium coefficients obtained. While quantification of the binding affinity is not explicitly shown in the figure, the weight score presented in the image corresponds to the software's quantitative assessment.
2. We have modified the term "lipid binding" to "translocation to LDs" to accurately describe the process.

Q5:

Thank you for your suggestion. We have replaced "small molecule 11094-59-0" with "Tridocosahexaenoin" and duly marked it in light blue in this revision.

^[1] Kozakov D, Hall DR, Xia B, Porter KA, Padhorny D, Yueh C, Beglov D, Vajda S. The ClusPro web server for protein-protein docking. Nat Protoc. 2017 Feb;12(2):255-278. doi: 10.1038/nprot.2016.169. Epub 2017 Jan 12. PMID: 28079879; PMCID: PMC5540229.

REVIEWERS' COMMENTS

Reviewer #2 (Remarks to the Author):

The authors have addressed my main concerns.